# Random Controlled Differential Equations

**Francesco Piatti, Thomas Cass & William F. Turner**
Department of Mathematics
Imperial College London

## Abstract

We introduce a training-efficient framework for time-series learning in which large randomly parameterized controlled and rough differential equations act as continuous-time reservoirs. These random dynamical systems map input paths to rich path-dependent representations, while only a linear readout layer is trained, yielding fast, scalable models with strong inductive bias. Building on this foundation, we propose two variants: (i) Random Fourier CDEs (RF-CDEs): these lift the input signal using random Fourier features prior to the dynamics, providing a kernel-free approximation of RBF-enhanced sequence models; (ii) Random Rough DEs (R-RDEs): these operate directly on rough-path inputs via a log-ODE discretisation, using log-signatures to capture higher-order temporal interactions while remaining stable and efficient. We prove that in the infinite-width limit, these models induce the RBF-lifted signature kernel and the rough signature kernel, respectively, offering a unified perspective on random-feature reservoirs, continuous-time deep architectures, and path-signature theory.

We evaluate both models across a range of time-series benchmarks, demonstrating competitive or superior performance. These methods provide a practical alternative to explicit signature computations, retaining their inductive bias while benefiting from the efficiency of random features. Code is publicly available at:
`https://github.com/FrancescoPiatti/RandomSigJax`

## 1 Introduction

Controlled differential equations (CDEs) generalize ordinary differential equations by allowing dynamics to be driven by an exogenous path $x : [0, T] \to \mathbb{R}^d$ rather than by time alone. This viewpoint has become central to modern time-series learning: it underlies the continuous-depth limit of residual networks (Cirone et al., 2023), connects naturally to deep state-space models and sequence models with long context (Rangapuram et al., 2018; Gu et al., 2021; Gu & Dao, 2023), and yields the *neural CDE* paradigm in which the vector field is represented by a neural network and learned from data (Kidger et al., 2020; Jhin et al., 2024). Beyond modelling, CDEs provide a clean analytical lens for studying expressivity and invariances of sequence models.

A complementary analytic tool for path-valued data is the *signature* of a path, i.e., the sequence of iterated integrals that linearizes CDE solution maps in the tensor algebra (Lyons, 1998). Signatures also induce powerful kernels on path space with universality and stability guarantees. In practice, however, signature features and signature kernels can be computationally demanding at high truncation levels, motivating approximations and random features.

Reservoir computing offers an appealing alternative: use a large, randomly initialized dynamical system to produce rich features of the input, and train only a linear readout (Lukoševičius & Jaeger, 2009). This training-light approach scales well and often works competitively. Yet, principled reservoirs for path data that come with nontrivial statistical or kernel limits – and that retain the continuous-time structure of CDEs – are less developed.

### 1.1 Related Literature

A growing body of research has drawn connections between random neural networks and kernel methods. Early works demonstrated that infinitely wide neural networks converge to Gaussian pro-

cesses (Neal, 1996; Williams, 1996). This idea was later extended to deep networks and their training dynamics: in the infinite-width limit, gradient descent on a network is equivalent to kernel regression with the so-called Neural Tangent Kernel (Jacot et al., 2018). In parallel, random feature models were introduced to approximate kernel maps with random projections (Rahimi & Recht, 2007; 2008). For example, random Fourier features approximate shift-invariant kernels with a linear readout, while Extreme Learning Machines fix hidden weights and train only the output layer, achieving universal approximation under standard conditions on the activation and sufficient width (Huang, 2014). These insights inspire reservoir computing, which leverages large random dynamical systems as feature extractors (Maass et al., 2002; Jaeger, 2007). This approach scales well and has demonstrated strong performance in time-series forecasting and classification tasks.

For sequence data, the path *signature* provides an expressive feature map with strong guarantees (Lyons, 1998), spawning a body of work on signature features and signature kernels in learning and statistics (Chevyrev & Kormilitzin, 2016; Cochrane et al., 2021; Salvi et al., 2021b; Toth & Oberhauser, 2020). A key advance is the PDE/Volterra characterisation and scalable computation of the untruncated signature kernel (Salvi et al., 2021a), with recent numerical refinements (Cass et al., 2025) and applications across regression, classification, and Bayesian inference (Lemercier et al., 2021a;b).

Randomized signatures compress path information by sampling random linear functionals of the (log-)signature coordinates. Discrete-time constructions with approximation and concentration guarantees were developed by Cuchiero et al. (2021), their practical efficacy as reservoirs for learning rough dynamics has been demonstrated in Compagnoni et al. (2023), while universal approximation on path space via finite mixtures of randomized signature features was established by Biagini et al. (2024).

Of particular relevance, Toth et al. (2025) propose *Random Fourier Signature Features* (RFSF): the input path is first lifted pointwise into a RBF reproducing kernel Hilbert space via random Fourier features and the signature transform is then approximated in that feature space. We adopt RFSF as a strong baseline and point of comparison for our random differential equation-based reservoirs.

## 1.2 CONTRIBUTIONS

In this work, we bridge these viewpoints by developing random feature models for path-valued data that leverage the continuous-time dynamics of CDEs. Our contributions can be summarized as follows:

- **Models.** We propose two architectures: (i) *RF-CDE*, which lifts inputs with random Fourier features and then evolves them through a random CDE; and (ii) *R-RDE*, which operates directly on geometric rough paths via a log-ODE discretization that uses log-signatures to capture higher-order temporal interactions.

- **Theory.** We prove infinite-width limits: RF-CDE converges to the RBF-lifted signature kernel, and R-RDE converges to the rough signature kernel; in both cases, the limiting feature maps induce Gaussian–process priors over path-functionals via the standard kernel–GP correspondence.

- **Efficiency.** At finite width, both models require training only a linear readout, yielding fast, scalable pipelines in the spirit of reservoir computing. We provide a user-friendly, optimized JAX implementation, `RandomSigJax`.

- **Experiments.** Across time-series benchmarks, our models are competitive with – or surpass – baselines, including Random Fourier Signature Features.

## 2 MATHEMATICAL BACKGROUND

We use the following notation throughout:

- $V$ and $W$ denote Banach spaces; $M_N(\mathbb{R})$ is the set of $N \times N$ real matrices; $C^1(J; V)$ is the space of continuously differentiable paths from $J \subset \mathbb{R}^+$ into $V$.

- $\Delta_T := \{(s, t) \in [0, T]^2 : 0 \leq s \leq t \leq T\}$ is the (time) two–simplex.

- $\xi_N$ is the Gaussian measure of matrices in $M_N(\mathbb{R})$: if the random matrix $A \sim \xi_N$ then its entries $A_{ij}$ are i.i.d. according to a standard normal distribution.

## 2.1 Controlled Differential Equations

Controlled differential equations (CDEs) describe dynamics driven by a path $x : [0, T] \to \mathbb{R}^d$ rather than by time alone:

$$Z_t = z_0 + \sum_{i=1}^{d} \int_0^t f_i(Z_s) \, dx_s^i, \qquad f_i : W \to W. \tag{1}$$

They capture how a system $Z_t \in W$ reacts to an external control $x$, and sit at the heart of continuous-time sequence models.

The main subtlety is *how to define the integrals* in Eq. 1. If $x$ has bounded variation – i.e. finite total variation, equivalently finite 1-variation (Definition A.1 in Appendix A) – the integrals are classical Riemann-Stieltjes integrals and the CDE is well-posed under standard Lipschitz conditions on $f_i$.

If $x$ is rougher but has finite $p$-variation with $p < 2$, Young integration applies and Eq. 1 still makes sense provided the vector fields are sufficiently regular. At the stochastic $p = 2$ boundary, the integrals may instead be understood in the Itô or Stratonovich sense. Beyond this threshold ($p > 2$), pathwise Riemann–Stieltjes/Young/stochastic integrals break down; we handle this regime by *lifting* $x$ to a (geometric) rough path carrying iterated integrals and interpret Eq. 1 via rough integration – see the rough-path background in Section 2.4.

## 2.2 Signature and Signature Kernels

A central tool in the analysis of controlled systems is the *path signature*, which encodes the essential information of a path through its iterated integrals. Let $x : [0, T] \to V$ be a continuous path of finite $p$-variation with $p < 2$. Then, for any $t \in [0, T]$, the signature $\mathrm{Sig}(x)_{0,t}$ is the unique solution to the *signature CDE*:

$$d \, \mathrm{Sig}(x)_{0,t} = \mathrm{Sig}(x)_{0,t} \otimes dx_t, \qquad \mathrm{Sig}(x)_0 = \mathbb{1} := (1, 0, 0, \dots),$$

taking values in the *tensor algebra*

$$T((V)) := \left\{ \mathbb{A} = (a^0, a^1, \dots) \,\middle|\, a^0 \in \mathbb{R}, \ a^k \in V^{\otimes k} \text{ for } k \geq 1 \right\}.$$

equipped with componentwise addition and tensor multiplication $\otimes$. Explicitly,

$$\mathrm{Sig}(x)_{0,T} = \left( 1, S^1(x), S^2(x), \dots \right), \quad S^k(x) = \int_{0 < t_1 < \cdots < t_k < T} dx_{t_1} \otimes \cdots \otimes dx_{t_k}.$$

The signature has several important properties, including: (i) it uniquely determines a path up to tree-like equivalence (Hambly & Lyons, 2010); (ii) robustness to missing or irregular samples; and (iii) *universality*, i.e. any continuous functional of a path can be approximated arbitrarily well by linear functionals of its signature.

As the signature is infinite dimensional, for practical use it is truncated. We define the *truncated tensor algebra* over $V$ of order $N \in \mathbb{N}$ as the quotient $T^N(V) := T((V))/T^{>N}$, by the ideal

$$T^{>N} = \{ \mathbb{A} = (a^0, a^1, \dots) \in T((V)) : a^0 = \cdots = a^N = 0 \},$$

and the truncated signature at level $N$ is $\mathrm{Sig}^N := \pi_{\leq N}(\mathrm{Sig}(\cdot))$, with $\pi_{\leq N}$ the canonical projection.

Further details on the tensor algebra are given in Appendix A.1, and Appendix A.2 reviews the main properties of the signature.

**Signature kernels.** Endowing $T((V))$ with a suitable inner product yields the *signature kernel*

$$K_{\mathrm{Sig}}^{x,y}(s, t) = \langle \mathrm{Sig}(x)_{0,s}, \mathrm{Sig}(y)_{0,t} \rangle_{T((V))}. \tag{2}$$

When the inner product on each tensor power $V^{\otimes k}$ is chosen to be the Hilbert-Schmidt inner product induced from $\langle \cdot, \cdot \rangle_V$, the resulting inner product on (a subsect of) $T((V))$ can be defined by linearity as $\langle v, w \rangle_{T((V))} = \sum_{k=0}^{\infty} \langle v_k, w_k \rangle_{V^{\otimes k}}$. The signature kernel is universal and, when the paths are differentiable, admits an alternative characterization as the solution to the linear hyperbolic PDE (Salvi et al., 2021a)

$$\partial_s \partial_t K_{\mathrm{Sig}}^{x,y}(s, t) = \langle \dot{x}_s, \dot{y}_t \rangle_V \, K_{\mathrm{Sig}}^{x,y}(s, t), \qquad K_{\mathrm{Sig}}^{x,y}(0, t) = K_{\mathrm{Sig}}^{x,y}(s, 0) = 1. \tag{3}$$

Further details on the signature kernels are provided in Appendix A.3.

## 2.3 RANDOM FOURIER SIGNATURE FEATURES

The RBF kernel admits an explicit feature representation: there is a map $\phi : \mathbb{R}^d \to \mathcal{H}$ into its reproducing kernel Hilbert space (RKHS) – Definition A.2 – such that $\kappa_{\text{RBF}}(x, y) = \langle \phi(x), \phi(y) \rangle_{\mathcal{H}}$. Equivalently, by *Bochner's theorem*, any shift-invariant positive-definite kernel $k(x, y) = f(x - y)$ can be written as

$$k(x, y) = \int_{\mathbb{R}^d} e^{i \, \omega^{\top} (x-y)} \, d\mu(\omega),$$

for a probability measure $\mu$ (Gaussian for RBF). This yields *Random Fourier Features (RFF)*: sample frequencies $\omega_1, \ldots, \omega_F \sim \mu$ and define the real map $\phi_{\mu}^F : \mathbb{R}^d \to \mathbb{R}^{2F}$

$$\phi_{\mu}^F(x) = \frac{1}{\sqrt{F}} \big( e^{i\langle \omega_1, x \rangle}, \ldots, e^{i\langle \omega_F, x \rangle} \big), \qquad \langle \phi_{\mu}^F(x), \phi_{\mu}^F(y) \rangle \approx \kappa_{\text{RBF}}(x, y). \tag{4}$$

where we concatenate the real and imaginary part. When the random Fourier feature map is applied along the path and then the signature is taken we are given the *naïve* Random Fourier Signature Features (RFSF), first proposed in Toth et al. (2025):

$$\text{RFSF}_{\text{N,F}}(x) := \text{Sig}^N(\phi_{\mu}^F(x)) \in T^N(\mathbb{R}^{2F}). \tag{5}$$

As $F$ grows, $\phi_{\mu}^F$ approximates the RKHS feature map of RBF; as $N$ grows, $\text{Sig}^N$ approaches the full signature. Hence taking the inner product $\langle \text{RFSF}_{\text{N,F}}(x), \text{RFSF}_{\text{N,F}}(y) \rangle_{T^N(\mathbb{R}^{2F})}$ provides a practical approximation to the *RBF–lifted signature kernel*, which can be defined in the limit as

$$K_{\text{Sig-RBF}}^{x,y}(s, t) \; = \; \langle \text{Sig}(\phi \circ x)_s, \; \text{Sig}(\phi \circ y)_t \rangle_{T((\mathcal{H}))}. \tag{6}$$

While computing the full signature suffers from the curse of dimensionality, Toth et al. (2025) develops projection schemes that render RFSFs computationally tractable. Recall that, from a computational standpoint, using explicit features rather than kernels often avoids constructing and inverting large Gram matrices, yielding far better scalability.

## 2.4 ROUGH PATHS

Rough path theory generalizes controlled differential equations to paths of limited regularity, including those with finite $p$-variation for $p > 2$. In contrast to classical paths, rough paths carry additional algebraic structure encoding iterated integrals, which enables a well-posed theory of integration and differential equations driven by such paths.

**Definition 2.1** (Rough Path). *Let $p \geq 1$ and let $\omega$ be a control (i.e. $\omega : \Delta_T \to [0, +\infty)$ is continuous, super-additive, and vanishes on the diagonal). A $p$-rough path over $V$ controlled by $\omega$ is a continuous map $\mathbb{X} : \Delta_T \to T^{\lfloor p \rfloor}(V)$ such that:*

1. *$\mathbb{X}_{s,t}^0 = 1$ for all $s, t \in \Delta_T$*

2. *Chen's identity holds: $\mathbb{X}_{s,u} \otimes \mathbb{X}_{u,t} = \mathbb{X}_{s,t}$*

3. *it has finite $p$-variation on $\Delta_T$ controlled by $\omega$, in the sense*

$$\|\pi_k(\mathbb{X})\|_{V^{\otimes k}} \leq \frac{\omega(s,t)^{i/p}}{\beta_p \Gamma(i/p)}, \quad \forall s, t \in \Delta_T, \quad \forall k = 1, \ldots, \lfloor p \rfloor,$$

*where $\beta_p \in \mathbb{R}$ is a constant that depends only on $p$, and the norm $\| \cdot \|_{V^{\otimes k}}$ is defined in Eq. 17 in Appendix A.*

**Definition 2.2** (Geometric rough path). *A $p$-rough path $\mathbb{X}$ is called* geometric *if there exists a sequence of bounded variation paths $(x^{(n)})_{n \geq 1}$ such that $\pi_{\lfloor p \rfloor}\big(\text{Sig}(x^{(n)})\big) \longrightarrow \mathbb{X}$ in the $p$-variation metric (see Definition A.3 in Appendix A).*

We denote by $\Omega_p(V)$ the space of $p$-rough paths and by $G\Omega_p(V) \subset \Omega_p(V)$ the geometric ones. Given our definitions $\Omega_1(\mathbb{R}^d)$ is the space of $d$-dimensional continuous paths of bounded variation.

The tensor algebra $T((V))$ carries the Lie bracket $[\mathbb{A}, \mathbb{B}] := \mathbb{A} \otimes \mathbb{B} - \mathbb{B} \otimes \mathbb{A}$. The *free Lie algebra* on $V$, denoted $\mathcal{L}((V))$, is the smallest Lie subalgebra of $T((V))$ containing $V$; elements are finite

linear combinations of iterated brackets $[e_{i_1}, [e_{i_2}, [\ldots, e_{i_k}] \cdots]]$, where $\{e_1, \ldots, e_d\}$ is a basis of $V$. Its degree-$m$ truncation is $\mathcal{L}^m(V) := \pi_{\leq m}(\mathcal{L}(V))$.

This allows us to define the *log-signature* as $\log(\mathrm{Sig}(x)) \in \mathcal{L}((V))$ (or $\log_n(\mathrm{Sig}(x)) \in \mathcal{L}^{\leq n}(V)$ at finite level), which collects only *Lie* monomials and thereby removes algebraic redundancies, yielding a more compact coordinate system than the raw signature. The logarithmic map acting on $T((V))$ is described by Eq. 18 in Appendix A.

For $p \geq 1$ and $q \geq 1$ real numbers, the *(rough)* signature kernel is the map defined for any two geometric $p$- and $q$-rough paths $\mathbb{X}, \mathbb{Y}$, by

$$K^{\mathbb{X},\mathbb{Y}}_{\mathrm{Sig}}(s,t) = \langle \mathrm{Sig}(\mathbb{X})_{0,s}, \mathrm{Sig}(\mathbb{Y})_{0,t}\rangle_{T((V))}. \tag{7}$$

Finally, rough paths can drive differential equations, which we use in this paper. Appendix A.5 provides a brief review, including the Universal Limit Theorem (Lyons, 1998), guaranteeing existence and uniqueness of RDEs driven by geometric rough paths under $\mathrm{Lip}(\gamma)$ vector fields (Definition A.5).

## 3 RANDOM CONTROLLED DIFFERENTIAL EQUATIONS

In this section, we review the random controlled differential equation model of Cirone et al. (2023), then introduce our variants and state the corresponding limit theorems.

### 3.1 R-CDE: RANDOM CONTROLLED DIFFERENTIAL EQUATIONS

Let $x \in C^1([0,T]; \mathbb{R}^d)$ and let $\mathcal{D}_M = \{0 = t_0 < \cdots < t_M = T\}$ be any partition of $[0,T]$. Consider a 1-layer, randomly initialized, homogeneous ResNet driven by $x$, with random readout $w \sim \xi_N$ (independent of the dynamics). Its output is

$$\Psi^N_\varphi(x) := \frac{1}{\sqrt{N}} \langle w, Z^N_{t_M}(x)\rangle_{\mathbb{R}^N},$$

where the hidden state evolves by the Euler-type recursion on $\mathcal{D}_M$

$$Z^N_{t_{i+1}}(x) = Z^N_{t_i}(x) + \frac{1}{\sqrt{N}} \sum_{k=1}^d A_k \, \varphi\big(Z^N_{t_i}(x)\big) \Delta x^k_{t_i}, \qquad Z^N_{t_0}(x) = z_0 \in \mathbb{R}^N,$$

with $\Delta x^k_{t_i} := x^k_{t_{i+1}} - x^k_{t_i}$, nonlinearity $\varphi$, and i.i.d. random matrices $A_k \sim \xi_N$ in $M_N(\mathbb{R})$.

Intuitively, as depth $M \to \infty$ (with mesh size $|\Delta_M| := \max_i(t_{i+1} - t_i) \to 0$), this recursion converges to a continuous-time controlled system. We take this limit as the definition of the *Random Controlled Differential Equation* (R-CDE):

$$dZ^N_t(x) = \frac{1}{\sqrt{N}} \sum_{i=1}^m A_i \, \varphi\big(Z^N_t(x)\big) dx^i_t, \qquad Z^N_0(x) = z_0 \in \mathbb{R}^N. \tag{8}$$

The expected inner product of these features converges to the signature kernel, and the readout converges to a Gaussian process with this covariance – thereby characterizing the joint infinite-width/continuous-depth limit of controlled ResNets.

**Theorem 3.1** (Cirone et al. (2023)). *Let $x, y \in C^1([0,T]; \mathbb{R}^d)$ and let $Z^N_s(x), Z^N_t(y)$ solve Eq. 8 with the same $(A_i)_{i=1}^d$ and $\varphi = \mathrm{id}$. Then for all $s, t \in [0,T]$,*

$$\lim_{N \to \infty} \frac{1}{N} \mathbb{E}_{\xi_N}\big[\langle Z^N_s(x), Z^N_t(y)\rangle_{\mathbb{R}^N}\big] = K^{x,y}_{\mathrm{sig}}(s,t),$$

*the (Hilbert–Schmidt) signature kernel of $(x,y)$, defined in Eq. 2. Moreover, with $w \sim \xi_N$ independent of $(A_i)$ and $Z^N(x)$,*

$$\lim_{N \to \infty} \Psi^N_\varphi(x) = \mathcal{GP}\big(0, K^{x,x}_{\mathrm{Sig}}\big),$$

*in the sense of finite-dimensional distributions.*

**Remark 3.1.** *Our theoretical analysis assumes Gaussian matrices for simplicity; however, the conclusions hold for any ensemble $\xi_N$ with the standard moment/tail conditions (centered, unit variance, sub-Gaussian operator–norm tails), as shown by Cass & Turner (2024).*

## 3.2 RF-CDE: RANDOM FOURIER CONTROLLED DIFFERENTIAL EQUATIONS

Motivated by the empirical success of Random Fourier Signature Features (Section 2.3) and by the RBF-lifted signature kernel (Eq. 6), we extend the R-CDE framework by incorporating a random Fourier lift of the driving signal.

Let $\phi_\mu^F : \mathbb{R}^d \to \mathbb{R}^{2F}$ be the RFF map in Eq. 4 and, for $x_t \in C^1([0,T], \mathbb{R}^d)$, we denote the lifted path by

$$X_t^F := \phi_\mu^F(x_t) \in \mathbb{R}^{2F},$$

where $\mu$ is the Gaussian measure. Notice that $X_t^F$ is also differentiable as it is a composition of differentiable functions. Then we define the *Random Fourier CDE* (RF-CDE) as

$$dZ_t^{N,F}(x) = \frac{1}{\sqrt{N}} \sum_{i=1}^{2F} A_i\, \varphi\big(Z_t^{N,F}(x)\big)\, dX_t^{F,i}, \qquad Z_0^{N,F}(x) = z_0 \in \mathbb{R}^N, \qquad (9)$$

where $z_0, (A_i) \sim \xi_N$ independent across $i$ and from the RFF randomness, and $X_t^{F,i}$ denotes the $i$-th component of the lifted path.

**Theorem 3.2.** *Let $x_t, y_t$ be differentiable paths on $[0,T]$ and $Z_s^{N,F}(x), Z_t^{N,F}(y)$ solve Eq. 9 with $\varphi = \mathrm{id}$ and the same $A_i \overset{i.i.d.}{\sim} \xi_N$ (independent of the RFF draw). Then, for every $s,t \in [0,T]$*

$$\lim_{F \to \infty} \lim_{N \to \infty} \frac{1}{N} \mathbb{E}_{\xi_N} \left[ \big\langle Z_s^{N,F}(x), Z_t^{N,F}(y) \big\rangle_{\mathbb{R}^N} \right] = K_{Sig\text{-}RBF}^{x,y}(s,t),$$

*where $K_{Sig\text{-}RBF}^{x,y}(s,t)$ denotes the RBF–lifted signature kernel (Eq. 6).*

We refer the reader to Appendix B.1 for the proof of this theorem.

**Gaussian–Process Interpretation.** With a fixed random reservoir and a trained linear readout, RF-CDE implements kernel ridge regression with kernel $N^{-1}\langle Z_s^{N,F}(x), Z_t^{N,F}(y)\rangle$, which converges to the RBF–lifted signature kernel as $N, F \to \infty$. In this limit, the RF-CDE reservoir defines a *Gaussian–process prior over path-functionals* with that signature-based covariance – mirroring the GP limit established for R-CDE in Section 3.1. This provides a clear interpretation of the model's inductive bias: RF-CDE inherits the expressive structure of signature kernels while retaining the scalability of random-feature reservoirs.

**Discretization.** In practice, we discretize Eq. 9, thereby extending its applicability beyond smooth drivers to piecewise-linear paths. We also include a bias vector $b_i \sim \xi_N$ together with scaling parameters $\sigma_A$, $\sigma_b$, and $\sigma_0$ (tuned via grid search). Applying an Euler scheme yields

$$\Delta Z_t^{N,F}(x) = \frac{1}{\sqrt{N}} \sum_{i=1}^{F} \Big( \sigma_A A_i\, \varphi\big(Z_t^{N,F}(x)\big) + \sigma_b b_i \Big) \Delta X_t^{F,i}, \qquad Z_0^{N,F}(x) = \sigma_0 z_0 \in \mathbb{R}^N, \quad (10)$$

where $\Delta X_t^{F,i} = X_{t_{i+1}}^{F,i} - X_{t_i}^{F,i}$ and $X_t^{F,i}$ is the $i$-th coordinate of the lifted path $X_t^F$.

## 3.3 R-RDE: RANDOM ROUGH DIFFERENTIAL EQUATIONS

We now extend the model to *non-smooth* drivers by working directly with geometric $p$-rough paths. This serves two purposes: (i) noisy time series often benefit from higher–order information, which signatures/log-signatures provide as stable features; (ii) in many applications the (log-)signature is already available or estimable, so operating *in rough-path space* avoids information loss.

Let $f \in \mathrm{Hom}(V, W)$ be a continuous linear map. For each $k \geq 1$, $f$ induces a map

$$f^{\otimes k} : V^{\otimes k} \to W^{\otimes k} \quad \text{s.t.} \quad f^{\otimes k}(v_1 \otimes \cdots \otimes v_k) := f(v_1) \otimes \cdots \otimes f(v_k) \quad \text{with} \quad f^{\otimes 0} := \mathrm{Id}.$$

The elements of $V^{\otimes k} \subset T((V))$ can be interpreted as functions on words of length $k$ over an alphabet $\mathcal{A}_d = 1, \ldots, d$. A *word* $w = i_1 i_2 \ldots i_k$ corresponds to the basis element $e_{i_1} \otimes \cdots \otimes e_{i_k}$ in $V^{\otimes k}$. Denote as $\mathcal{W}_d^m$ the set of all words formed by letters in $\mathcal{A}_d$ of length $|w| \leq m$, and

$W_d := \bigcup_{m \geq 0} \mathcal{W}_d^m$. Let $(A_i) \in \mathrm{End}(\mathbb{R}^N)$ (the algebra of endomorphisms of $\mathbb{R}^N$ under composition with unit $\mathrm{Id}_N$), and let $\Gamma_A : T((\mathbb{R}^d)) \to \mathrm{End}(\mathbb{R}^N)$ be the unital algebra homomorphism

$$\Gamma_A(\mathbb{G}) = \sum_{w \in \mathcal{W}_d} A_w \langle \mathbb{G}, w \rangle, \qquad \text{where } A_w := \frac{1}{N^{\frac{k}{2}}} A_{i_1} \cdots A_{i_k} \quad \text{for } w = i_1 \cdots i_k \in \mathcal{W}_d, \quad (11)$$

extended multiplicatively by $\Gamma_A(uv) = \Gamma_A(u) \circ \Gamma_A(v)$ and $\Gamma_A(1) = \mathrm{Id}_N$. Here $\langle \mathbb{G}, w \rangle$ denotes the $w$–coordinate of the tensor $\mathbb{G} \in T((\mathbb{R}^d))$. The map $\Gamma_A$ is well defined on group-like tensors $G(\mathbb{R}^d) \subset T((\mathbb{R}^d))$ (see Appendix A.4) and in particular on signature increments of geometric rough paths; we refer to Lemma 6 in Appendix B.2 for the precise statement and proof.

**Linear development along a path.** For a bounded-variation path $x : [0, T] \to \mathbb{R}^d$ set

$$S_t^A(x) := \Gamma_A\big(\mathrm{Sig}(x)_{0,t}\big) \in \mathrm{End}(\mathbb{R}^N), \tag{12}$$

**Lemma 1.** *Let $\Gamma_A$ be as in Eq. 11. If $x : [0, T] \to \mathbb{R}^d$ has bounded variation and $x_t^A := \sum_{i=1}^d A_i x_t^i$, with $A_i \in \mathrm{End}(\mathbb{R}^N)$, then $S_t^A(x)$ in Eq. 12 is the unique solution of the linear CDE*

$$dS_t^A(x) = S_t^A(x) \circ dx_t^A, \qquad S_0^A(x) = \mathrm{Id}_N \in \mathrm{End}(\mathbb{R}^N). \tag{13}$$

See Appendix B.2 for the proof.

By continuity of the Itô–Lyons map (Theorem A.5), Eq. 13 extends to geometric rough drivers. If $\mathbb{X} \in G\Omega_p(\mathbb{R}^d)$, there exists a unique matrix–valued geometric $p$-rough path $\mathbb{S}^A \in G\Omega_p(\mathrm{End}(\mathbb{R}^N))$ given by the *canonical* lift of the solution to the rough linear equation

$$dS_t^A(\mathbb{X}) = S_t^A(\mathbb{X}) \circ d\mathbb{X}_t \qquad S_0^A(x) = \mathrm{Id}_N \in \mathrm{End}(\mathbb{R}^N), \tag{14}$$

i.e the first level of the lift: $\pi_1\big(\mathbb{S}_{0,t}^A\big) := S_t^A(\mathbb{X}) \in \mathrm{End}(\mathbb{R}^N)$.

**Random RDE features.** Let $(A_i)_{i=1}^d \overset{\text{i.i.d.}}{\sim} \xi_N$ be Gaussian random matrices in $M_N(\mathbb{R})$, and let $\mathbb{S}^A$ be the matrix–valued geometric $p$-rough path associated with Eq. 14. Define the *one–form*

$$f : \mathbb{R}^N \longrightarrow \mathrm{Hom}(\mathrm{End}(\mathbb{R}^N), \mathbb{R}^N), \qquad f(z)[M] := M\big(\varphi(z)\big),$$

with non-linearity $\varphi \in \mathrm{Lip}(\gamma)$ (Definition A.5) and $\gamma > p$. The random-feature path $Z^N(\mathbb{X}) : [0, T] \to \mathbb{R}^N$ is then defined as the unique solution of the Random RDE

$$dZ_t^N(\mathbb{X}) = f\big(Z_t^N\big) d\mathbb{S}_t^A, \qquad Z_0^N = z_0 \in \mathbb{R}^N, \tag{15}$$

where $z_0 \sim \xi_N$ is independent of $\{A_i\}_{i=1}^d$.

**Remark 3.2.** *In the smooth case (so $\mathbb{X} \equiv \mathrm{Sig}(x)$ with $x$ of bounded variation), Eq. 15 yields $dZ_t = \sum_{i=1}^d \big(S_t^A(x) A_i \varphi(Z_t)\big) dx_t^i$. A brief derivation is provided in Appendix B.2.*

We now state and prove two theorems: the first establishes existence and uniqueness, while the second shows convergence to the rough signature kernel introduced in Section 2.2.

**Theorem 3.3** (Existence and uniqueness). *Let $\mathbb{X} \in G\Omega_p(\mathbb{R}^d)$ with $p \geq 1$ and $\varphi \in \mathrm{Lip}(\gamma)$ with $\gamma > p$. Then the R-RDE 15 admits a unique solution $Z^N \in C([0, T]; \mathbb{R}^N)$, and the Itô–Lyons map $(\mathbb{X}, z_0) \mapsto Z^N$ is continuous in the rough-path topology.*

*Proof.* This is a direct application of the Universal Limit Theorem (Theorem A.5) under $\mathrm{Lip}(\gamma)$ vector fields (Definition A.5) with $\gamma > p$.

**Theorem 3.4.** *Let $\mathbb{X} \in G\Omega_p(\mathbb{R}^d)$ and $\mathbb{Y} \in G\Omega_q(\mathbb{R}^d)$ be geometric rough paths. Let $Z_s^N(\mathbb{X})$ and $Z_t^N(\mathbb{Y})$ be the solutions of Eq. 15 with $\varphi = \mathrm{id}$ and the same matrices $\{A_i\}_{i=1}^d$ (with $A_i \sim \xi_N$ i.i.d.). Then for all $s, t \in [0, T]$,*

$$\lim_{N \to \infty} \frac{1}{N} \mathbb{E}_{\xi_N}\big[\langle Z_s^N(\mathbb{X}), Z_t^N(\mathbb{Y})\rangle_{\mathbb{R}^N}\big] = K_{\mathrm{Sig}}^{\mathbb{X},\mathbb{Y}}(s, t),$$

*where where $K_{\mathrm{Sig}}^{\mathbb{X},\mathbb{Y}}$ denotes the rough signature kernel defined in Eq. 7.*

We refer the reader to Appendix B.3 for the proof of this theorem.

**Gaussian–process interpretation.** Analogously to the RCDE and RF-CDE settings, a fixed R-RDE reservoir with a trained linear readout induces a Gaussian–process prior over path–functionals, with covariance given by the rough signature kernel $K_{\text{Sig}}^{\mathbb{X},\mathbb{Y}}$.

**Log–ODE discretization.** For rough drivers, naïve Euler schemes ignore the algebraic structure of the signal, breaking Chen's multiplicativity. The *log–ODE method* (Lyons, 2014) addresses this by summarizing each time step $[t_i, t_{i+1}]$ via the *log-signature*

$$\mathfrak{L}_i := \log_m\left(\mathbb{X}_{t_i, t_{i+1}}\right) \in \mathcal{L}^m(\mathbb{R}^d),$$

which maps increments into the step-$m$ free Lie algebra – see Section 2.4. One then advances the state on $[t_i, t_{i+1}]$ by solving an ODE with *constant Lie coefficients* $\mathfrak{L}_i$. This preserves the group/Chen structure exactly and removes the algebraic redundancies of the tensor algebra.

Let $\widetilde{\mathcal{W}}_d^m$ be a fixed Hall/Lyndon basis of Lie words of length $|w| \leq m$. Given a collection of random matrices $\{B_1, \ldots, B_d\}$ with $B_i \overset{\text{i.i.d.}}{\sim} \xi_N$, define the linear map $\Pi_B : \mathcal{L}^m(\mathbb{R}^d) \to \text{End}(\mathbb{R}^N)$ by

$$\Pi_B(\mathbb{G}) := \sum_{w \in \widetilde{\mathcal{W}}_d^m} B^{(w)} \left\langle \log_m(\mathbb{G}_{t_i, t_{i+1}}), w \right\rangle, \qquad B^{(w)} := \frac{1}{N^{\frac{k}{2}}}[\cdots[[B_{i_1}, B_{i_2}], B_{i_3}], \ldots, B_{i_k}]$$

for $w = i_1 \cdots i_k$, where $[A, B] = AB - BA$ is the Lie bracket.

Given $\mathbb{X} \in G\Omega_m(\mathbb{R}^d)$ and a partition $\mathcal{D}_M = \{0 = t_0 < \cdots < t_M = T\}$, we update

$$\widetilde{Z}_{t_{i+1}}^N = \widetilde{Z}_{t_i}^N + \Pi_B(\mathbb{X}_{t_i, t_{i+1}})\,\varphi\big(\widetilde{Z}_{t_i}^N\big), \qquad \widetilde{Z}_{t_0}^N = z_0 \in \mathbb{R}^N, \tag{16}$$

which uses only the log-signature coefficients $\langle \log_m(\mathbb{X}_{t_i, t_{i+1}}), w \rangle$ and the corresponding commutators $B^{(w)}$, keeping the discretization faithful to the rough-path algebra while remaining explicit. Scaling and bias hyperparameters are incorporated analogously to the RF-CDE discretization (Eq. 10) and tuned by grid search.

While more computationally expensive than RF–CDE, R–RDE can be advantageous on very long sequences: the log-ODE discretisation operates on log-signatures computed over chunks, thereby shortening the effective trajectory length fed to the random differential equation.

## 4 EXPERIMENTS

In this section, we detail our random differential equation models' performance for time series classification. We implement RF-CDE via the discretization in Eq. 10 and R-RDE via the log-ODE scheme in Eq. 16. For completeness, we also benchmark the R-CDE of Cirone et al. (2023) – described in Section 2.1 – which, to our knowledge, has not been tested.

**Benchmarks.** We compare against Random Fourier Signature Features (RFSF) in the two projection variants of Toth et al. (2025) – Diagonal Projection (DP) and Tensorized Random Projection (TRP). We also benchmark the PDE-based signature kernel (SigPDE) with the RBF base kernel (Salvi et al., 2021a), and standard time-series baselines: Random Fourier Features (RFF), RBF, GAK, and Random Warping Series (RWS). For space, full results for RBF, GAK, RWS, and RFF are deferred to Appendix C, as these methods rarely attain state-of-the-art performance on our suite but are included for completeness. We do not include the truncated signature kernel (Király & Oberhauser, 2019) in our benchmarks, as its exponential feature explosion makes it impractical in all but small and low-dimensional datasets. Neural benchmark such as Neural CDE (Kidger et al., 2020) and Neural RDE (Morrill et al., 2021) have also been included in the ablation study of UEA time series classification and in the Hurst-exponent classification experiment.

**Remark 4.1.** *SigPDE, GAK, and RBF are not random-feature methods: in the SVM setting they require computing and inverting the kernel Gram matrix, which can be a bottleneck as the number of samples grows. By contrast, random-feature models learn only a linear readout on top of fixed random dynamics, avoiding kernel matrix operations and retaining linear-in-samples complexity.*

**Computational Time.** Table 4 summarizes the computational complexity of our signature-based random-feature extractors. These models scale linearly with the sequence length $\ell$, unlike kernel baselines such as SigPDE, RBF, and GAK which scale quadratically. Among our methods, R-RDE is typically the slowest due to an additional $O(N^3)$ component arising from matrix development; however, this cubic term is independent of the batch size $B$, so it can be precomputed.

| R-CDE | RFCDE (ours) | R-RDE (ours) | RFSF-DP | RFSF-TRP |
|:---:|:---:|:---:|:---:|:---:|
| $O(B\ell N^2 d)$ | $O(B\ell F(N^2 + d))$ | $O(N^2 d^M(B\ell + N))$ | $O(B\ell F(Md + 2^M))$ | $O(B\ell M(dF + F^2))$ |

Table 1: Asymptotic feature-extraction cost (ignoring the final linear readout). Here $B$ is batch size, $F$ the number of random Fourier features, $d$ the input dimension, $\ell$ the sequence length, $M$ the truncation level of the signature, and $N$ the (output) feature dimension.

## 4.1 UEA TIME SERIES CLASSIFICATION

**UEA Datasets.** The UEA archive (Dau et al., 2019) is a collection of datasets for benchmarking classifiers on multivariate time series classification problems. The data modality ranges from various sources e.g. human activity recognition, motion and ECG classification, audio spectra recognition, and others. A summary of the dataset characteristics can be found in Table 2 in Dau et al. (2019).

**Experimental Setup.** Full details, including the grid-search ranges, are deferred to Appendix C.1.

**Results.** With 250 random features, RF-CDE is the strongest of random-feature models on average, followed by the non-parametric SigPDE baseline. Averaging across the 16 UEA datasets, RF-CDE attains 0.741 accuracy versus 0.738 for SigPDE, while RFSF variants are slightly behind (0.725–0.726), and R-RDE and R-CDE trail further (0.708 and 0.695, respectively). RF-CDE is particularly competitive on medium-difficulty tasks (e.g., *Libras*, *NATOPS*), and R-RDE occasionally leads among random-feature methods on structure-rich datasets (e.g., *UWaveGestureLibrary*). These trends suggest that injecting an RBF lift before the controlled dynamics (RF-CDE) is an effective way to capture local geometry in continuous time, whereas the rough-path variant (R-RDE) helps when higher-order interactions matter. Table 4.1 reports the results.

| | R-CDE | RF-CDE | R-RDE | RFSF-DP | RFSF-TRP | SigPDE |
|---|:---:|:---:|:---:|:---:|:---:|:---:|
| ArticularyWordRecognition | 0.950 | 0.967 | 0.957 | 0.977 | **0.983** | **0.983** |
| AtrialFibrillation | 0.333 | **0.467** | **0.467** | 0.400 | 0.266 | 0.333 |
| BasicMotions | **1.000** | **1.000** | **1.000** | 0.975 | 0.975 | **1.000** |
| Cricket | **0.972** | **0.972** | 0.902 | **0.972** | 0.958 | **0.972** |
| EigenWorms | 0.420 | 0.630 | 0.612 | 0.786 | 0.771 | **0.794** |
| Epilepsy | 0.935 | **0.971** | 0.935 | 0.942 | 0.942 | 0.891 |
| EthanolConcentration | 0.312 | 0.358 | 0.373 | 0.430 | 0.407 | **0.460** |
| FingerMovements | 0.550 | 0.550 | 0.530 | 0.570 | 0.530 | **0.610** |
| Handwriting | 0.331 | 0.331 | 0.331 | 0.380 | 0.362 | **0.409** |
| Libras | 0.889 | **0.911** | 0.867 | 0.833 | 0.906 | 0.867 |
| NATOPS | 0.872 | **0.944** | 0.906 | 0.889 | 0.906 | 0.928 |
| RacketSports | 0.809 | 0.809 | 0.737 | 0.829 | 0.809 | **0.849** |
| SelfRegulationSCP1 | 0.877 | 0.877 | 0.840 | 0.887 | **0.904** | **0.904** |
| SelfRegulationSCP2 | 0.555 | 0.555 | **0.567** | 0.483 | 0.494 | 0.544 |
| StandWalkJump | 0.467 | **0.667** | 0.400 | 0.400 | 0.533 | 0.400 |
| UWaveGestureLibrary | 0.853 | 0.844 | **0.903** | 0.856 | 0.853 | 0.866 |
| Avg. acc. ($\uparrow$) | 0.695 | **0.741** | 0.708 | 0.726 | 0.725 | 0.738 |
| Avg. rank ($\downarrow$) | 4.250 | 3.062 | 4.125 | 3.406 | 3.594 | **2.562** |

Table 2: UEA test accuracies with $N = 250$ signature-based random-feature models (SigPDE is a kernel baseline: no random features). For each row, the best result is highlighted in **bold**.

**Ablation: Number of Features.** We repeat the full protocol with 64 and 500 random features. In the low–budget setting, RF-CDE performs particularly well relative to the RFSF and neural baselines. On the other hand, doubling the feature budget yields modest gains – typically a few percentage points on the more challenging datasets – while leaving easier tasks essentially unchanged. Being a kernel methods, SigPDE is unaffected by this ablation. Results are included in Appendix C.

## 4.2 CLASSIFICATION OF ROUGH SIGNALS VIA HURST EXPONENT RECOVERY

To further assess the ability of our model to extract fine-grained geometric information from irregular time-series data, we introduce a controlled classification task based on synthetic fractional Brownian motion (fBm). Each sample consists of fBm with Hurst exponent $H \in \{0.05, 0.15, \ldots, 0.75\}$, and the goal is to correctly identify the underlying Hurst parameter from the observed path. Recall that $H$ controls the roughness of the process through its $p$-variation.

We evaluate two variants of the task. **V1** uses the raw fBm trajectories. **V2** applies per-sample standardisation (zero mean and unit variance), thereby forcing the models to exploit only geometric features and long-range dependence. Full details of this experiment are deferred to Appendix C.5. Table 3 reports classification accuracies across a range of competing baselines. In both settings, our R-RDE model achieves consistently strong performance, and in the most challenging regime (**V2**, $N = 64$), it preserves a clear performance margin over alternative approaches.

Table 3: Hurst–exponent classification accuracy. Here $N$ denotes the feature dimension of the random-feature models (with neural baselines matched in parameter count).

| Setting | R-CDE | RF-CDE | R-RDE | RFSF-DP | RFSF-TRP | NCDE | NRDE |
|---|---|---|---|---|---|---|---|
| **V1** – $N = 64$ | 0.870 | 0.895 | **0.955** | 0.840 | 0.895 | 0.905 | 0.920 |
| **V1** – $N = 100$ | 0.900 | 0.945 | **0.950** | 0.890 | 0.910 | 0.895 | 0.945 |
| **V2** – $N = 64$ | 0.635 | 0.645 | **0.735** | 0.630 | 0.650 | 0.650 | 0.675 |
| **V2** – $N = 100$ | 0.650 | 0.695 | **0.730** | 0.675 | 0.675 | 0.650 | 0.685 |

## 4.3 ROBUSTNESS TO MISSING DATA

To assess the robustness of our models to incomplete observations, we perform an additional experiment on multivariate time series from the UEA archive. We synthetically introduce missing data by randomly removing individual time points along the test trajectories. The experiment is described in Appendix C.4, and the corresponding accuracy tables are provided therein (Tables 7 and 8). The results show that our models remains competitive even under substantial information loss: in particular, RF–CDE exhibits the most stable performance as the missingness level increases.

## 5 CONCLUSIONS

We introduced a training-efficient framework for time-series learning based on random continuous-time reservoirs whose infinite-width limits coincide with established path kernels: RF-CDE yields the RBF-lifted signature kernel, and R-RDE yields the rough signature kernel. This places our models on the same framework as infinite-width neural networks. Empirically, with only a few hundred features, both models are competitive on UEA benchmarks while avoiding kernel-matrix inversion and scaling linearly in sequence length. The result is a scalable alternative to explicit signature computation.

**Future Directions.** Looking forward, we see natural extensions: learn (or sparsify) the spectral measures that define the reservoirs, and couple our continuous-time features with probabilistic heads for calibrated uncertainty and streaming inference. It will also be interesting to study NTK dynamics around the random reservoir, to design adaptive log-structured discretizations for very long contexts.

## ACKNOWLEDGMENTS

TC has been supported in part by UK Research and Innovation (UKRI) through the Engineering and Physical Sciences Research Council (EPSRC) Programme Grant: UKRI1010 - *High order mathematical and computational infrastructure for streamed data that enhance contemporary generative and large language models*. FP and WFT have been supported by the EPSRC Centre for Doctoral Training in Mathematics of Random Systems: Analysis, Modelling and Simulation (EP/S023925/1). ChatGPT-5 was used by FP to refine the clarity and style of selected paragraphs. For the purpose of open access, the authors have applied a Creative Commons Attribution (CC BY) licence to any Author Accepted Manuscript version arising.

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

# A  ALGEBRAIC AND ANALYTIC BACKGROUND

This appendix collects the algebraic objects (tensor and free Lie algebras, group-like elements), the analytic objects (signatures, rough paths), and the two key theorems we rely on (the PDE characterization of the signature kernel and Lyons' Universal Limit Theorem). We keep statements self-contained; detailed proofs can be found in standard references (Cass & Salvi, 2024; Lyons, 1998).

## A.1  TENSOR ALGEBRA

Let $V$ be a Banach space. The spaces of formal polynomials and formal power series over $V$ are defined respectively as

$$T(V) = \bigoplus_{k=0}^{\infty} V^{\otimes k}, \qquad \text{and} \qquad T((V)) = \prod_{k=0}^{\infty} V^{\otimes k},$$

where $V^{\otimes k}$ denotes the $k$-fold tensor product of $V$. Both $T(V)$ and $T((V))$ can be endowed with the operations of component-wise addition and multiplication $\otimes$, the latter defined for any two elements $\mathbb{A} = (a_0, a_1, \dots)$ and $\mathbb{B} = (b_0, b_1, \dots)$ as

$$\mathbb{A} \otimes \mathbb{B} = (c_0, c_1, c_2, \dots), \qquad \text{where} \quad V^{\otimes k} \ni c_k = \sum_{i=0}^{k} a_i \otimes b_{k-i}, \ \forall k \geq 0.$$

When endowed with these two operations and the natural action of $\mathbb{R}$ by $\lambda \mathbb{A} = (\lambda a_0, \lambda a_1, \dots)$, $T((V))$ becomes a real, non-commutative unital algebra with unit $1 = (1, 0, 0, \dots)$ called the *tensor algebra*.

The *level-$m$ truncated tensor algebra* over $V$ of order $m \in \mathbb{N}$ is the quotient

$$T^m(V) := T((V))/T^{>m}(V) \cong \bigoplus_{k=0}^{m} V^{\otimes k},$$

where

$$T^{>m}(V) := \{\mathbb{A} = (a^0, a^1, \dots) \in T((V)) : a^0 = \cdots = a^m = 0\}$$

We denote by $\pi_{\leq m} : T((V)) \to T^m(V)$ the canonical projection and by $\pi_k : T((V)) \to V^{\otimes k}$ the level maps.

Finally, we can define a norm on $V^{\otimes k}$ as

$$\|v\|_{V^{\otimes k}} = \sqrt{\prod_{i=1}^{k} \langle v_i, v_i \rangle_V}, \qquad \text{for } v = v_i \dots v_k. \tag{17}$$

## A.2  SIGNATURES AND THEIR BASIC PROPERTIES

Signatures extend iterated integrals beyond smooth curves, but to do so we must quantify how "rough" a path is. The right scale is *p-variation*: it measures the cumulative oscillation of a path and yields a topology under which Young integrals (for $p < 2$) and, more generally, rough integrals (for $p \geq 2$) are well posed.

**Definition A.1** (*p*-variation). *Let $x : [0, T] \to V$ be continuous. For any $[s, t] \subseteq [0, T]$,*

$$\|x\|_{p-\mathrm{var},[s,t]} = \left( \sup_{\mathcal{D} \subset [s,t]} \sum_{t_i \in \mathcal{D}} \left\| x_{t_{i+1}} - x_{t_i} \right\|^p \right)^{1/p},$$

*where the supremum is over all partitions $\mathcal{D}$ of $[s, t]$.*

The induced *p-variation metric* on $C([0, T]; V)$ is

$$d_{p\text{-var}}(x, y) := \|x - y\|_{p\text{-var},[0,T]}.$$

Let $x : [0, T] \to V$ be a path of bounded variation. Its *signature* over $[s, t]$ is

$$\text{Sig}(x)_{s,t} = \left(1, S^1(x), S^2(x), \dots\right), \quad S^k(x) = \int_{s < u_1 < \cdots < u_k < t} dx_{u_1} \otimes \cdots \otimes dx_{u_k}.$$

The signature satisfies *Chen's identity* (it is multiplicative):

$$\text{Sig}(x)_{s,u} \otimes \text{Sig}(x)_{u,t} = \text{Sig}(x)_{s,t}$$

**Lemma 2** (Factorial decay). *For $x$ of bounded variation and all $k \geq 1$,*

$$\left\| \int \cdots \int_{s < u_1 < \cdots < u_k < t} dx_{u_1} \otimes \cdots \otimes dx_{u_k} \right\|_{V^{\otimes k}} \leq \frac{(\|x\|_{1\text{-}var;[s,t]})^k}{k!}.$$

*where the norm on $V^{\otimes k}$ is given by Eq. 17 and $\| \cdot \|_{1\text{-}var;[s,t]}$ denotes the $1$-variation norm given by Definition A.1.*

Three foundational properties make signatures effective for learning:

**Theorem A.1** (Uniqueness up to tree-like equivalence). *If $\text{Sig}(x)_{0,T} = \text{Sig}(y)_{0,T}$ then $x$ and $y$ are tree-like equivalent; conversely, tree-like equivalent paths have equal signatures. On classes where tree-like structure is ruled out (e.g. reduced paths), the signature is injective.*

**Theorem A.2** (Universality of linear functionals on signatures, (Hambly & Lyons, 2010)). *Let $\mathcal{K}$ be a compact set of bounded-variation paths (modulo tree-like equivalence). Then the linear span of coordinate iterated integrals $\{\langle \ell, \text{Sig}(\cdot)_{0,T} \rangle : \ell \in T((V)) \text{ finite}\}$ is dense in $C(\mathcal{K})$, the space of continuous functions on $\mathcal{K}$ with the topology of uniform convergence.*

**Lemma 3** (Reparameterization invariance). *Let $x : [t_0, T] \to \mathbb{R}^d$ be a continuous path of bounded variation and let $[a, b]$ and $[c, d]$ be two subintervals of $[t_0, T]$. Let $\lambda : [c, d] \to [a, b]$ be a reparameterization. Then $\text{Sig}(x)_{a,b} = S(x \circ \lambda)_{c,d}$.*

For some applications it might be important to keep the time parameterization of the path $x$. In this case, it suffices to add time as an extra coordinate of $x$ to get the time-augmented path $\widehat{x} : t \mapsto (t, x_t)$.

### A.3 SIGNATURE KERNELS AND THE PDE CHARACTERIZATION

Kernel methods measure similarity via inner products in a (possibly infinite-dimensional) feature space. This is formalized by the notion of a *reproducing kernel Hilbert space* (RKHS): a Hilbert space of functions in which point evaluation is a continuous linear functional represented by the kernel. We recall the standard definition below.

**Definition A.2.** *Let $\mathcal{X}$ be a nonempty set and $k$ be a positive definite kernel on $\mathcal{X}$. A Hilbert space $\mathcal{H}_k$ of real-valued functions on $\mathcal{X}$ equipped with an inner product $\langle \cdot, \cdot \rangle_{\mathcal{H}_k}$ is called a reproducing kernel Hilbert space (RKHS) with reproducing kernel $k$, if for any $x \in \mathcal{X}$ and for any $f \in \mathcal{H}_k$ the following two conditions are satisfied:*

1. *the feature map $k(x, \cdot) \in \mathcal{H}_k$;*

2. *the reproducing property $f(x) = \langle f, k(x, \cdot) \rangle_{\mathcal{H}_k}$ holds.*

Intuitively, the element $k(x, \cdot) \in \mathcal{H}_k$ plays the role of an (often infinite-dimensional) *feature vector* for $x$. An immediate consequence of the reproducing property is the feature–space inner product identity

$$k(x, y) = \langle k(x, \cdot), k(y, \cdot) \rangle_{\mathcal{H}_k}, \qquad x, y \in \mathcal{X}.$$

In our setting, the feature map of a path is its *signature* – the sequence of iterated integrals living in the tensor algebra. Given an (Hilbert-Schmidt) inner product $\langle \cdot, \cdot \rangle_V$ on $V$, define for $k \geq 1$

$$\left\langle v_1 \otimes \cdots \otimes v_k, \ w_1 \otimes \cdots \otimes w_k \right\rangle_{V^{\otimes k}} := \prod_{j=1}^{k} \langle v_j, w_j \rangle_V.$$

For $\mathbb{A} = (a_0, a_1, \ldots)$ and $\mathbb{B} = (b_0, b_1, \ldots)$ in $T((V))$, set

$$\langle \mathbb{A}, \mathbb{B} \rangle_{T((V))} := \sum_{k=0}^{\infty} \langle a_k, b_k \rangle_{V^{\otimes k}}.$$

This induces the *signature kernel*

$$K_{\mathrm{Sig}}^{x,y}(s,t) := \big\langle \mathrm{Sig}(x)_{0,s}, \, \mathrm{Sig}(y)_{0,t} \big\rangle_{T((V))}.$$

When $x, y$ are differentiable, $K_{\mathrm{Sig}}^{x,y}$ is characterized as the unique solution to a linear hyperbolic Goursat PDE.

**Theorem A.3** (PDE/Volterra characterization, (Salvi et al., 2021a)). *Let $x, y \in C^1([0,T]; V)$. Then $k(s,t) := K_{\mathrm{Sig}}^{x,y}(s,t)$ solves*

$$\partial_s \partial_t k(s,t) = \langle \dot{x}_s, \dot{y}_t \rangle_V \, k(s,t), \qquad k(s,0) = k(0,t) = 1,$$

*equivalently,*

$$k(s,t) = 1 + \int_0^s \int_0^t \langle \dot{x}_u, \dot{y}_v \rangle_V \, k(u,v) \, dv \, du.$$

*Conversely, the (unique) solution of this problem coincides with $\langle \mathrm{Sig}(x)_{0,s}, \mathrm{Sig}(y)_{0,t} \rangle$.*

A direct corollary is a universality statement for the induced kernel on path space.

**Theorem A.4** (Universality/characteristicness of the signature kernel). *On compact sets of paths (modulo tree-like equivalence), the signature kernel is universal (its RKHS is dense in continuous functions) and characteristic (mean embeddings of Borel probability measures are injective).*

### A.4 ROUGH PATHS AND LIE ALGEBRAS

For tensor-valued paths, the $p$-variation extends in the usual rough-path way: apply $p$-variation level-wise to each homogeneous tensor component and combine (up to an equivalent norm) to obtain the standard $p$-variation metric on $T^{\lfloor p \rfloor}(V)$.

**Definition A.3** ($p$-variation metric). *The $p$-variation metric of two $p$-rough paths $\mathbb{X}, \mathbb{Y} : \Delta_T \to T^{\lfloor p \rfloor}(V)$ is defined as follows*

$$d_p(\mathbb{X}, \mathbb{Y}) = \max_{1 \leq k \leq \lfloor p \rfloor} \sup_{\mathcal{D}} \left( \sum_{t_k \in \mathcal{D}} \big\| \pi_k(\mathbb{X}_{t_i, t_{i+1}}) - \pi_k(\mathbb{Y}_{t_i, t_{i+1}}) \big\|_{V^{\otimes k}}^{p/k} \right)^{1/p},$$

*where the supremum is taken over all partitions $\mathcal{D}$ of the interval $[0, T]$, and the norm on $V^{\otimes k}$ is given by Eq. 17.*

This metric is the natural topology for rough paths (Definition 2.1): the class of *geometric $p$-rough paths* $G\Omega_p(V)$ is the closure, under $d_p$, of truncated signatures of bounded-variation paths (Definition 2.2).

The associative tensor algebra $(T((V)), \otimes)$ carries a commutator

$$[\mathbb{A}, \mathbb{B}] = \mathbb{A} \otimes \mathbb{B} - \mathbb{B} \otimes \mathbb{A}.$$

Equipped with $[\cdot, \cdot]$, the same underlying vector space becomes a (noncommutative) Lie algebra. Iterated brackets quantify non-commutativity and generate the "Lie words" that will organize the algebraic content of iterated integrals.

**Definition A.4** (Lie polynomials and Lie series). *Let $L_0 = 0$, $L_1 = V$, and $L_{k+1} = [V, L_k]$, with $[V, U]$ denoting the linear span of all elements of the form $[e, f]$ where $(e, f) \in V \times U$ for any two linear subspaces $V, U$ of $T((V))$.*

*The space of Lie polynomials over $V$, denoted as $\mathcal{L}(V)$, is defined as:*

$$\mathcal{L}(V) = \bigoplus_{k=0}^{\infty} L_k.$$

*The space of Lie formal series over $V$, denoted as $\mathcal{L}((V)) \subset T((V))$ is defined as*

$$\mathcal{L}((V)) = \left\{ \mathbb{L} = \left( l^0, l^1, \ldots \right) \mid \forall k \geq 0, l^k \in L_k \right\}.$$

For any $n \geq 1$, *the step-$n$ free Lie algebra* is defined by $\mathcal{L}^n(V) := \pi_{\leq n}(\mathcal{L}((V)))$ with elements called Lie polynomials of degree $n$. Define the (formal) exponential and logarithm w.r.t. tensor multiplication,

$$\exp(\mathbb{A}) := \sum_{n=0}^{\infty} \frac{\mathbb{A}^{\otimes n}}{n!}, \qquad \log(\mathbb{1} + \mathbb{A}) := \sum_{n=1}^{\infty} \frac{(-1)^{n-1}}{n} \mathbb{A}^{\otimes n}, \tag{18}$$

and their level-$m$ truncations $\exp_m := \pi_{\leq m} \circ \exp$, $\log_m := \pi_{\leq m} \circ \log$.

The *group-like* subset

$$G(V) := \exp\left( \mathcal{L}((V)) \right) \subset T((V))$$

is a Lie group under the tensor product; at level $n$ we write $G^n(V) := \pi_{\leq n}(G(V))$ with mutually inverse maps

$$\exp : \mathcal{L}((V)) \to G(V) \qquad \text{and} \qquad \log : G(V) \to \mathcal{L}((V)),$$

and

$$\exp_n : \mathcal{L}^n(V) \to G^n(V) \qquad \text{and} \qquad \log_n : G^n(V) \to \mathcal{L}^n(V).$$

For any bounded-variation path (and, by continuity, for geometric rough paths), the signature $\mathrm{Sig}(x)_{s,t} \in G(V)$ is *group-like*. Correspondingly, the *log-signature* $\log(\mathrm{Sig}(x)_{s,t})$ lies in the free Lie algebra $\mathcal{L}((V))$ (or its truncation $\mathcal{L}^n(V)$ at finite step). This identification is what allows Lie-algebraic discretizations (e.g. log-ODE schemes) that respect the path's multiplicative structure.

A.5 ROUGH DIFFERENTIAL EQUATIONS (RDEs)

This section follows Cass & Salvi (2024) and gives a precise pathwise meaning to rough differential equations (RDEs)

$$dY_t = f(Y_t) \, d\mathbb{X}_t, \quad Y_0 = y_0 \in \mathbb{R}^N \tag{19}$$

driven by a geometric rough path $\mathbb{X}$ (Definition 2.2).

The main idea is to approximate the (geometric) rough path $\mathbb{X} \in G\Omega_p(V)$ by signatures of smooth/bounded–variation paths in $p$-variation, solve the ordinary controlled differential equations (CDEs) along those smooth drivers, and define the rough solution as the uniform limit of the smooth solutions. The key tool is Lyons' Universal Limit Theorem (Theorem A.5), which also yields stability/continuity of the solution map (the Itô –Lyons map).

Before stating it we recall the definition of $\gamma$-Lipschitz function (in the sense of Stein).

**Definition A.5** (Lip($\gamma$) functions)**.** *Let $V, W$ be two normed space and let $\gamma > 0$. A function $g : V \to W$ is called $\gamma$-Lipschitz if $g$ is $\lfloor \gamma \rfloor$ times continuously differentiable and such that there exists a constant $M \geq 0$ such that the supremum norm of its $k^{th}$ derivative, $k = 0, \ldots, \lfloor \gamma \rfloor$, and the $(\gamma - \lfloor \gamma \rfloor)$-Hölder norm of its $\lfloor \gamma \rfloor^{th}$ derivative are bounded by $M$. The smallest $M$ satisfying these conditions is the $\gamma$ Lipschitz norm of $g$, denoted by $\|g\|_{Lip \, \gamma} := \|g\|_{Lip \, \gamma(V,W)}$. We denote by $Lip \, \gamma(V, W)$ the space of $\gamma$-Lipschitz functions from $V$ to $W$.*

**One-forms and the rough integral (informal).** In an RDE the map $f : W \to \mathrm{Hom}(V, W)$ is naturally viewed as a *one-form*: at each state $y \in W$, $f(y)$ is a linear map $V \to W$ to be integrated against the (rough) increment of the driver. When $V = \mathbb{R}^d$ we often write $f = (f_1, \ldots, f_d)$ with $f_i : W \to W$, so that the coordinate form is

$$dY_t = \sum_{i=1}^{d} f_i(Y_t) \, dX_t^i.$$

Since $\mathbb{X}$ is rough, the integral cannot be defined by Riemann–Stieltjes sums at level 1 only. Instead one uses all available iterated integrals of $\mathbb{X}$ up to level $m = \lfloor p \rfloor$.

**Controlled paths and compensated Riemann sums.** A path $Y : [0, T] \to W$ is *controlled by* $\mathbb{X}$ if its increments admit an expansion $Y_{u,v} = Y'_u \mathbb{X}^1_{u,v} + R_{u,v}$ with higher-order consistency and suitable $p$-variation bounds on the remainder $R_{u,v}$. For such $Y$ and $f \in \mathrm{Lip}(\gamma)$ with $\gamma > p$, the rough integral $\int_0^t f(Y_u) \, d\mathbb{X}_u$ is defined as the limit, as $|\mathcal{D}| \to 0$, of the compensated sums

$$\sum_{[u,v] \in \mathcal{D}} \left( f(Y_u) \, \mathbb{X}^1_{u,v} + Df(Y_u)[f(Y_u)] \, \mathbb{X}^2_{u,v} + \cdots + D^{m-1} f(Y_u) [\underbrace{f(Y_u), \ldots, f(Y_u)}_{m-1 \text{ times}}] \, \mathbb{X}^m_{u,v} \right),$$

where $\mathbb{X}^k_{u,v} \in V^{\otimes k}$ are the level-$k$ increments of the geometric rough path. This construction agrees with the classical Riemann–Stieltjes (or Young) integral when the driver is smooth (or has $p < 2$), and it is the pathwise notion used in Eq. 19.

**Theorem A.5** (Universal Limit Theorem (Lyons, 1998)). *Let $p \geq 1$ and let $X^n : [0, T] \to V$ be a sequence of continuous paths of bounded variation which converges in $p$-variation to a geometric $p$-rough path $\mathbb{X} : \Delta_T \to T^{\lfloor p \rfloor}(V)$. Let $f : V \to \mathrm{Hom}(V, W)$ be a $\mathrm{Lip}(\gamma)$ function with $\gamma > p$. Consider the controlled differential equations*

$$dY^n_t = f(Y^n_t) dX^n_t, \qquad Y^n_0 = y_0 \in W \tag{20}$$

*Then, there exists a unique geometric rough path $\mathbb{Z} = (\mathbb{X}, \mathbb{Y}) : \Delta_T \to T^{\lfloor p \rfloor}(V \oplus W)$ such that $Y^n$ converges to $\mathbb{Y}$ in $p$-variation. Moreover, the Itô map $I_f : (y_0, \mathbb{X}) \to \mathbb{Y}$ is continuous in $p$-variation.*

**Definition A.6** (RDE solution). *Let $\mathbb{X} \in G\Omega_p(V)$ be a geometric $p$-rough path. We say that the continuous path $Y : [0, T] \to W$ of finite $p$-variation is a solution to the RDE*

$$dY_t = f(Y_t) \, d\mathbb{X}_t, \quad Y_0 = y_0 \in W$$

*if $Y$ belongs to the set of (uniform) limit points constructed in Theorem A.5. In particular, if $f : W \to \mathrm{Hom}(V, W)$ is linear or $\gamma$-Lipschitz with $\gamma > p$, then $Y$ is unique.*

The notion of RDE solution presented in Definition A.6 maps a geometric $p$-rough path to a $W$-valued continuous path of finite $p$-variation. However, it might be desirable to construct a "full" solution also as a geometric rough path. This is the case, for example, if one is interested in using a solution to a first RDE to be the driving signal for a second RDE. More precisely, we will say that $\mathbb{Y} \in G\Omega(W)$ is the (full) solution to the RDE

$$d\mathbb{Y}_t = f(\mathbb{Y}_t) \, d\mathbb{X}_t, \quad \text{started at} \quad \mathbb{Y}_0 \in \mathrm{Sig}^{\lfloor p \rfloor}(\Omega_1(W))$$

if there exists a sequence $(X^n)$ of continuous bounded variation paths such that the sequence of truncated signatures $(\mathrm{Sig}^{\lfloor p \rfloor}(X^n))$ converges in $p$-variation to $\mathbb{X}$ and such that the sequence $(\mathbb{Y}_0 \cdot \mathrm{Sig}^{\lfloor p \rfloor}(Y^n))$ converges uniformly on $[0, T]$ to $\mathbb{Y}$ as $n \to \infty$, where $\{Y^n\}$ are the solutions to the CDEs 20, with $Y_0 = \pi_1(\mathbb{Y}_0)$.

## B PROOFS

Before proving the main theorems, we record two auxiliary results used in our proofs. First, we show that the inner product of time-derivatives of Random Fourier Feature lifts converges almost surely to the corresponding RKHS cross term; we include a self-contained proof. Second, we invoke a trace–moment identity for products of random matrices from Cass & Turner (2024) (proof therein).

**Lemma 4.** *Let* $x, y \in C^1([0, T], \mathbb{R}^d)$ *and* $\phi^F : \mathbb{R}^d \to \mathbb{R}^{2F}$ *be the random Fourier feature map*

$$\phi_\mu^F(z) := \frac{1}{\sqrt{F}} \Big( \cos(\omega_1^\top z), \ \sin(\omega_1^\top z), \ \ldots, \ \cos(\omega_F^\top z), \ \sin(\omega_F^\top z) \Big) \in \mathbb{R}^{2F},$$

*where* $\{\omega_j\}_{j=1}^F \overset{i.i.d.}{\sim} \mu$ *and* $\mu$ *is the standard Gaussian measure on* $\mathbb{R}^d$. *Define the lifted curves*

$$X_t^F := \phi_\mu^F(x_t), \qquad and \qquad Y_t^F := \phi_\mu^F(y_t).$$

*Let* $\mathcal{H} := L^2(\mu; \mathbb{R}^2)$ *and define the (infinite-dimensional) feature map*

$$\phi(z) := \big( \cos(\omega^\top z), \ \sin(\omega^\top z) \big) \in \mathbb{R}^2, \qquad \langle u, v \rangle_\mathcal{H} := \int_{\mathbb{R}^d} u(\omega)^\top v(\omega) \, d\mu(\omega).$$

*Let* $X_t := \phi(x_t) \in \mathcal{H}$, *and* $Y_t := \phi(y_t) \in \mathcal{H}$; *then, for every* $s, t \in [0, T]$, *we have*

1. *almost sure convergence*

$$\big\langle \dot{X}_s^F, \ \dot{Y}_t^F \big\rangle_{\mathbb{R}^{2F}} \xrightarrow[F \to \infty]{a.s.} \big\langle \dot{X}_s, \ \dot{Y}_t \big\rangle_\mathcal{H}. \tag{21}$$

2. *convergence in* $L^1$:

$$\int_0^T \int_0^T \big| \big\langle \dot{X}_s^F, \ \dot{Y}_t^F \big\rangle_{\mathbb{R}^{2F}} - \big\langle \dot{X}_s, \ \dot{Y}_t \big\rangle_\mathcal{H} \big| \, dt \, ds \xrightarrow{F \to \infty} 0. \tag{22}$$

*where* $\dot{X}_s^F$ *is the derivative w.r.t. time of* $X_s^F$, *and similarly for* $\dot{Y}_t^F$, $\dot{X}_s$, *and* $\dot{Y}_t$.

**Proof.** Differentiating component-wise,

$$\frac{d}{dt} \cos(\omega_j^\top x_t) = -\sin(\omega_j^\top x_t) \, \omega_j^\top \dot{x}_t, \qquad \frac{d}{dt} \sin(\omega_j^\top x_t) = \cos(\omega_j^\top x_t) \, \omega_j^\top \dot{x}_t,$$

and similarly with $x_t, \dot{x}_t$ replaced by $y_t, \dot{y}_t$. Hence

$$\dot{X}_s^F = \frac{1}{\sqrt{F}} \Big( -\sin(\omega_j^\top x_s) \, \omega_j^\top \dot{x}_s, \ \cos(\omega_j^\top x_s) \, \omega_j^\top \dot{x}_s \Big)_{j=1}^F,$$

and analogously for $\dot{Y}_t^F$. The Euclidean inner product becomes

$$\big\langle \dot{X}_s^F, \dot{Y}_t^F \big\rangle_{\mathbb{R}^{2F}} = \frac{1}{F} \sum_{j=1}^F (\omega_j^\top \dot{x}_s)(\omega_j^\top \dot{y}_t) \Big( \sin(\omega_j^\top x_s) \sin(\omega_j^\top y_t) + \cos(\omega_j^\top x_s) \cos(\omega_j^\top y_t) \Big)$$

$$= \frac{1}{F} \sum_{j=1}^F (\omega_j^\top \dot{x}_s)(\omega_j^\top \dot{y}_t) \, \cos \big( \omega_j^\top (x_s - y_t) \big) =: \frac{1}{F} \sum_{j=1}^F g_{\omega_j}(s, t).$$

Define

$$g_\omega(s, t) := (\omega^\top \dot{x}_s)(\omega^\top \dot{y}_t) \, \cos \big( \omega^\top (x_s - y_t) \big).$$

Then $\{g_{\omega_j}(s, t)\}_{j=1}^F$ are i.i.d. with mean

$$\mathbb{E}_\mu[g_\omega(s, t)] = \int_{\mathbb{R}^d} (\omega^\top \dot{x}_s)(\omega^\top \dot{y}_t) \, \cos \big( \omega^\top (x_s - y_t) \big) \, d\mu(\omega).$$

On the other hand, in the RKHS model $\mathcal{H} = L^2(\mu; \mathbb{R}^2)$,

$$\dot{X}_s(\omega) = \big( -\sin(\omega^\top x_s)\, \omega^\top \dot{x}_s,\ \cos(\omega^\top x_s)\, \omega^\top \dot{x}_s \big),$$

and similarly for $\dot{Y}_t(\omega)$. Therefore

$$\langle \dot{X}_s, \dot{Y}_t \rangle_{\mathcal{H}} = \int_{\mathbb{R}^d} (\omega^\top \dot{x}_s)(\omega^\top \dot{y}_t) \Big( \sin(\omega^\top x_s) \sin(\omega^\top y_t) + \cos(\omega^\top x_s) \cos(\omega^\top y_t) \Big)\, d\mu(\omega)$$

$$= \int_{\mathbb{R}^d} (\omega^\top \dot{x}_s)(\omega^\top \dot{y}_t)\, \cos\big( \omega^\top (x_s - y_t) \big)\, d\mu(\omega) = \mathbb{E}_\mu[g_\omega(s,t)].$$

Thus $\mathbb{E}_\mu[g_\omega(s,t)] = \langle \dot{X}_s, \dot{Y}_t \rangle_{\mathcal{H}}$. By Cauchy–Schwarz and $|\cos| \le 1$

$$|g_\omega(s,t)| \le \|\omega\|^2\, \|\dot{x}_s\|\, \|\dot{y}_t\|.$$

Since

  i. the sequence $\{g_{\omega_j}(s,t)\}_{j \ge 1}$ is i.i.d.,
  ii. $x, y \in C^1([0,T], \mathbb{R}^d)$ implies that $\|\dot{x}_s\|, \|\dot{y}_t\|$ are bounded on $[0,T]$,
  iii. $\int \|\omega\|^2\, d\mu(\omega) < \infty$ as the Gaussian distribution has finite variance, so $g_\omega(s,t)$ is integrable,

we can apply the strong law of large numbers, giving that

$$\frac{1}{F} \sum_{j=1}^F g_{\omega_j}(s,t) \xrightarrow[F \to \infty]{\text{a.s.}} \mathbb{E}_\mu[g_\omega(s,t)] = \langle \dot{X}_s, \dot{Y}_t \rangle_{\mathcal{H}},$$

which proves Eq. 21.

For the $L^1$ convergence,

$$\big| \langle \dot{X}_s^F, \dot{Y}_t^F \rangle_{\mathbb{R}^{2F}} \big| \le \Big( \frac{1}{F} \sum_{j=1}^F \|\omega_j\|^2 \Big) \|\dot{x}_u\|\, \|\dot{y}_v\| \xrightarrow[F \to \infty]{\text{a.s.}} \mathbb{E}_\mu \|\omega\|^2\, \|\dot{x}_u\|\, \|\dot{y}_v\|.$$

Since $\mathbb{E}_\mu \|\omega\|^2 < \infty$ for Gaussian $\mu$, there exists $C(\omega) < \infty$ such that, for all $F$,

$$|\langle \dot{X}_s^F, \dot{Y}_t^F \rangle_{\mathbb{R}^{2F}}| \le C(\omega)\, \|\dot{x}_s\|\, \|\dot{y}_t\|, \qquad \text{and} \qquad |\langle \dot{X}_s, \dot{Y}_t \rangle_{\mathcal{H}}| \le C(\omega)\, \|\dot{x}_s\|\, \|\dot{y}_t\|,$$

Hence,

$$\big| \langle \dot{X}_s^F, \dot{Y}_t^F \rangle_{\mathbb{R}^{2F}} - \langle \dot{X}_s, \dot{Y}_t \rangle_{\mathcal{H}} \big| \le 2C(\omega)\, \|\dot{x}_s\|\, \|\dot{y}_t\| \tag{23}$$

Since $\dot{x}, \dot{y} \in L^1([0,T])$, the envelope on the right is integrable over $[0,T]^2$. Combining the almost-sure pointwise convergence with the bound in Eq. 23, and using the continuity (hence measurability) of the maps

$$(s,t) \mapsto \langle \dot{X}_s^F, \dot{Y}_t^F \rangle_{\mathbb{R}^{2F}} \qquad \text{and} \qquad (s,t) \mapsto \langle \dot{X}_s, \dot{Y}_t \rangle_{\mathcal{H}}$$

the hypotheses of the dominated convergence theorem are satisfied. Therefore,

$$\int_0^T \int_0^T \big| \langle \dot{X}_s^F, \dot{Y}_t^F \rangle_{\mathbb{R}^{2F}} - \langle \dot{X}_s, \dot{Y}_t \rangle_{\mathcal{H}} \big|\, dt\, ds \xrightarrow{F \to \infty} 0,$$

which proves Eq. 22.

$\square$

**Lemma 5** (Cass & Turner (2024)). *Suppose that for each $N \in \mathbb{N}$, $\{A_i^N : 1 \leq i \leq d\}$ is a collection of $d$ Gaussian matrices. Let $n, m \geq 0$ be non-negative integers and consider the words $\mathbf{I} = i_1 \ldots i_n \in \mathcal{W}_d^n$, and $\mathbf{J} = j_1 \ldots j_m \in \mathcal{W}_d^m$ (where $\mathcal{W}_d$ is defined in Section 3.3), with corresponding matrix products*

$$A_{\mathbf{I}}^N := A_{i_1}^N \ldots A_{i_n}^N$$

$$A_{\mathbf{I} \star \mathbf{J}}^N := \left(A_{\mathbf{I}}^N\right)^T A_{\mathbf{J}}^N$$

*Then, setting $k = n + m$,*

$$\lim_{N \to \infty} \frac{1}{N^{\frac{k}{2}+1}} \mathbb{E}\left[\mathrm{tr}\left(A_{\mathbf{I} \star \mathbf{J}}^N\right)\right] = \begin{cases} 1, & \text{if } \mathbf{I} = \mathbf{J} \\ 0, & \text{otherwise} \end{cases}$$

*We use the convention that if $n = 0$, then $A_{\mathbf{I}}^N = \mathrm{Id}_N$, the $N \times N$ identity matrix.*

### B.1 PROOF OF THEOREM 3.2

For convenience we restate Theorem 3.2 and provide its proof.

**Theorem.** *Let $x_t, y_t$ be differentiable paths on $[0, T]$ and $Z_s^{N,F}(x), Z_t^{N,F}(y)$ solve Eq. 9 with $\varphi = \mathrm{id}$ and the same $A_i \overset{i.i.d.}{\sim} \xi_N$ (independent of the RFF draw). Then, for every $s, t \in [0, T]$*

$$\lim_{F \to \infty} \lim_{N \to \infty} \frac{1}{N} \mathbb{E}_{\xi_N}\left[\langle Z_s^{N,F}(x), Z_t^{N,F}(y)\rangle_{\mathbb{R}^N}\right] = K_{\text{Sig-RBF}}^{x,y}(s, t),$$

*where $K_{\text{Sig-RBF}}^{x,y}(s, t)$ denotes the RBF–lifted signature kernel (Eq. 6).*

**Proof.** Recall from Section 3.2 that we denote

$$X_t^F := \phi_\mu^F(x_t) \in \mathbb{R}^{2F}, \qquad \text{and} \qquad Y_t^F := \phi_\mu^F(y_t) \in \mathbb{R}^{2F}.$$

where $\phi_\mu^F$ is the random Fourier map defined in Eq. 4.

For fixed $F$, the R-CDE limit (Theorem 3.1) applied to the drivers $X^F$ and $Y^F$ gives

$$\lim_{N \to \infty} \frac{1}{N} \mathbb{E}_{\xi_N}\left[\langle Z_s^{N,F}(x), Z_t^{N,F}(y)\rangle_{\mathbb{R}^F}\right] = k_F(s, t) \qquad \text{where} \qquad k_F := K_{\text{sig}}^{X^F, Y^F}, \qquad (24)$$

Let $k := K_{\text{Sig-RBF}}^{x,y}$. By the PDE/Volterra characterization of the signature kernel (Theorem A.3), $k_F, k : [0, T]^2 \to \mathbb{R}$ are the unique solutions of the two-parameter Volterra equations

$$k_F(s, t) = 1 + \int_0^s \int_0^t q_F(u, v) \, k_F(u, v) \, dv \, du,$$

and

$$k(s, t) = 1 + \int_0^s \int_0^t q(u, v) \, k(u, v) \, dv \, du,$$

respectively, with driving kernels $q_F(u, v) := \langle \dot{X}_u^F, \dot{Y}_v^F \rangle_{\mathbb{R}^{2F}}$ and $q(u, v) := \langle \dot{X}_u, \dot{Y}_v \rangle_{\mathcal{H}}$, where $\mathcal{H}$ is the RKHS of the RBF feature map.

By Lemma 4, $q_F \to q$ in $L^1([0, T]^2)$ almost surely (over the RFF draw), and there exists an envelope

$$|\langle \dot{X}_s^F, \dot{Y}_t^F \rangle_{\mathbb{R}^{2F}}| \leq C(\omega) \|\dot{x}_s\| \|\dot{y}_t\|, \qquad \text{and} \qquad |\langle \dot{X}_s, \dot{Y}_t \rangle_{\mathcal{H}}| \leq C(\omega) \|\dot{x}_s\| \|\dot{y}_t\|,$$

with $\|\dot{x}_u\|, \|\dot{y}_v\| \in L^1(0, T)$ and $C(\omega) < \infty$. The standard Volterra stability estimate via the two-parameter Grönwall inequality (Defranco, 1976), therefore yields

$$\|k_F - k\|_\infty \leq \exp\left(C\|\dot{x}_s\|_{L^1} \|\dot{y}_t\|_{L^1}\right) \|q_F - q\|_{L^1}.$$

Consequently, $k_F(s,t) \to k(s,t) = K_{\text{Sig-RBF}}^{x,y}(s,t)$ uniformly on $[0,T]^2$ for $\mu$-almost every RFF draw. Taking $F \to \infty$ in Eq. 24 and using this uniform convergence (together with the uniform bound implied by the envelope in the stability estimate) proves the theorem by dominated convergence.

$\square$

## B.2 PROOFS AND LEMMAS FOR SECTION 3.3

We first state and prove a lemma ensuring that $\Gamma_A$ is well defined and convergent on group-like elements (signature increments). We then prove Lemma 1 from the main text. Finally, we include a short derivation of the smooth–driver setting mentioned in Remark 3.2.

**Lemma 6** (Absolute convergence of $\Gamma_A$ on signature increments). *Let $p \geq 1$, $d \in \mathbb{N}$, and let $\mathbb{X} \in G\Omega_p(\mathbb{R}^d)$ with control $\omega$. Fix matrices $A_1, \ldots, A_d \in \text{End}(\mathbb{R}^N)$ and set*

$$\kappa := \max_{1 \leq i \leq d} \frac{1}{\sqrt{N}} \|A_i\| < \infty.$$

*For a word $w = i_1 \cdots i_k$ define $A_w := N^{-\frac{k}{2}} A_{i_1} \cdots A_{i_k}$ and $\Gamma_A$ by Eq. 11. Then for every $(s,t) \in \Delta_T$ the series*

$$\sum_{w \in \mathcal{W}_d} A_w \langle \mathbb{X}_{s,t}, w \rangle$$

*converges absolutely in $\text{End}(\mathbb{R}^N)$. Consequently $\Gamma_A(\mathbb{X}_{s,t})$ is well-defined and $(s,t) \mapsto \Gamma_A(\mathbb{X}_{s,t})$ is continuous.*

By sub-multiplicativity and the definition of $\kappa$,

$$\|A_w\| = N^{-\frac{k}{2}} \|A_{i_1} \cdots A_{i_k}\| \leq N^{-\frac{k}{2}} \prod_{j=1}^{k} \|A_{i_j}\| \leq \kappa^k \quad \text{for } |w| = k.$$

Group the series by word length and use the triangle inequality:

$$\sum_{w \in \mathcal{W}_d} \|A_w\| |\langle \mathbb{X}_{s,t}, w \rangle| \leq \sum_{k=0}^{\infty} \kappa^k \sum_{|w|=k} |\langle \mathbb{X}_{s,t}, w \rangle|.$$

By Cauchy–Schwarz,

$$\sum_{|w|=k} |\langle \mathbb{X}_{s,t}, w \rangle| \leq d^{\frac{k}{2}} \|\pi_k(\mathbb{X}_{s,t})\|,$$

where $\|\cdot\|$ is the Hilbert–Schmidt norm. Geometric $p$-rough paths satisfy the factorial decay (see Definition 2.1):

$$\|\pi_k(\mathbb{X}_{s,t})\| \leq \frac{C_p \, \omega(s,t)^{\frac{k}{p}}}{\Gamma(\frac{k}{p}+1)}$$

for some $C_p > 0$ that depends only on $p$. Hence

$$\sum_{w \in \mathcal{W}_d} \|A_w\| |\langle \mathbb{X}_{s,t}, w \rangle| \leq C_p \sum_{k=0}^{\infty} \left( \kappa \, d^{1/2} \right)^k \frac{\omega(s,t)^{\frac{k}{p}}}{\Gamma(\frac{k}{p}+1)}.$$

By Stirling's formula, $\Gamma(k/p+1) \sim (k/p)^{k/p} e^{-k/p} \sqrt{2\pi k/p}$, which outgrows any exponential; thus the right-hand series converges for all $\omega(s,t) < \infty$. Absolute convergence implies the claim.

$\square$

### B.2.1 PROOF OF LEMMA 1

For convenience we restate Theorem 1 and provide its proof.

**Lemma.** *Let $\Gamma_A$ be as in Eq. 11. If $x : [0,T] \to \mathbb{R}^d$ has bounded variation and $x_t^A := \sum_{i=1}^d A_i\,x_t^i$, with $A_i \in \mathrm{End}(\mathbb{R}^N)$, then $S_t^A(x)$ in Eq. 12 is the unique solution of the linear matrix CDE*

$$dS_t^A(x) = S_t^A(x) \circ dx_t^A, \qquad S_0^A(x) = \mathrm{Id}_N \in \mathrm{End}(\mathbb{R}^N).$$

**Proof.** For $x$ of bounded variation, the signature solves Chen's integral equation

$$\mathrm{Sig}(x)_{s,t} = \mathbb{1} + \int_{(s,t]} \mathrm{Sig}(x)_{s,u} \otimes dx_u,$$

with $dx_u = (dx_u^1, \ldots, dx_u^d)$. Applying the algebra homomorphism $\Gamma_A$ (which sends $1 \mapsto I_N$, concatenation $\otimes$ to composition, and $e_i \mapsto A_i$) yields

$$S_{s,t}^A(x) := \Gamma_A\big(\mathrm{Sig}(x)_{s,t}\big) = \mathrm{Id}_N + \int_{(s,t]} S_{s,u}^A(x) \circ dx_u^A, \qquad dx_u^A := \sum_{i=1}^d A_i\,dx_u^i,$$

i.e. $dS_t^A(x) = S_t^A(x) \circ dx_t^A$ with $S_0^A(x) = \mathrm{Id}_N \in \mathrm{End}(\mathbb{R}^N)$. Uniqueness follows from standard Picard iteration for linear matrix CDEs driven by bounded-variation paths.

$\square$

### B.2.2 REMARK 3.2 (SMOOTH-DRIVER DERIVATION)

Assume $x$ has bounded variation so that $\mathbb{X} = \mathrm{Sig}(x)$. Starting from

$$Z_t = Z_0 + \int_0^t dS_u^A\big(\varphi(Z_u)\big),$$

expand $dS_u^A$ using $S_u^A = \sum_{w \in \mathcal{W}_d} A_w \langle \mathrm{Sig}(x)_{0,u}, w \rangle$:

$$Z_t = Z_0 + \int_0^t \Big( \sum_{w \in \mathcal{W}_d} A_w\,d\langle \mathrm{Sig}(x)_{0,u}, w \rangle \Big)\big(\varphi(Z_u)\big).$$

Write each non-empty word as $w = \widehat{w}\,i$ (last letter $i$) and use $d\langle \mathrm{Sig}(x)_{0,u}, \widehat{w}i \rangle = \langle \mathrm{Sig}(x)_{0,u}, \widehat{w} \rangle\,dx_u^i$ and $d\langle \mathrm{Sig}(x)_{0,u}, \emptyset \rangle = 0$ to get

$$Z_t = Z_0 + \sum_{i=1}^d \int_0^t \Big( \sum_{\widehat{w} \in \mathcal{W}_d} A_{\widehat{w}i}\,\langle \mathrm{Sig}(x)_{0,u}, \widehat{w} \rangle \Big)\big(\varphi(Z_u)\big)\,dx_u^i.$$

By multiplicativity of $\Gamma_A$ (so $A_{\widehat{w}i} = A_{\widehat{w}} A_i$),

$$\sum_{\widehat{w}} A_{\widehat{w}i}\,\langle \mathrm{Sig}(x)_{0,u}, \widehat{w} \rangle = \Big( \sum_{\widehat{w}} A_{\widehat{w}}\,\langle \mathrm{Sig}(x)_{0,u}, \widehat{w} \rangle \Big) A_i = S_u^A A_i,$$

and hence

$$Z_t = Z_0 + \sum_{i=1}^d \int_0^t \big(S_u^A A_i\,\varphi(Z_u)\big)\,dx_u^i.$$

### B.3 PROOF OF THEOREM 3.4

For convenience we restate Theorem 3.4 and provide its proof.

**Theorem.** *Let $\mathbb{X} \in G\Omega_p(\mathbb{R}^d)$ and $\mathbb{Y} \in G\Omega_q(\mathbb{R}^d)$ be geometric rough paths. Let $Z_s^N(\mathbb{X})$ and $Z_t^N(\mathbb{Y})$ be the solutions of Eq. 15 with $\varphi = \mathrm{id}$ and the same matrices $\{A_i\}_{i=1}^d$ (with $A_i \sim \xi_N$ i.i.d.). Then for all $s, t \in [0, T]$,*

$$\lim_{N \to \infty} \frac{1}{N} \mathbb{E}_{\xi_N} \left[ \langle Z_s^N(\mathbb{X}), Z_t^N(\mathbb{Y}) \rangle_{\mathbb{R}^N} \right] = K_{\mathrm{Sig}}^{\mathbb{X}, \mathbb{Y}}(s, t),$$

*where where $K_{\mathrm{Sig}}^{\mathbb{X}, \mathbb{Y}}$ denotes the rough signature kernel defined in Eq. 7.*

**Proof.** As in the smooth/CDE case, the (matrix) Dyson/Chen expansion under $\varphi = \mathrm{id}$ gives

$$\lim_{N \to \infty} \frac{1}{N} \mathbb{E}_{\xi_N} \left[ \langle Z_s^N(\mathbb{X}), Z_t^N(\mathbb{Y}) \rangle_{\mathcal{H}} \right] = \lim_{N \to \infty} \mathbb{E}_{\xi_N} \left[ \sum_{\mathbf{I}, \mathbf{J} \in \mathcal{W}_d}^{\infty} \frac{1}{N^{\frac{|\mathbf{I} \star \mathbf{J}|}{2} + 1}} \mathrm{tr} \left( A_{\mathbf{I} \star \mathbf{J}}^N \right) \mathrm{Sig}^{\mathbf{I}}(\mathbb{X})_s \, \mathrm{Sig}^{\mathbf{J}}(\mathbb{Y})_t \right]$$

where $\mathcal{W}_d$ is introduced in Section 3.3 and $\mathrm{Sig}^{\mathbf{I}}(\cdot)$ denotes the coordinate of the signature associated to the word $\mathbf{I}$ (i.e. $\mathrm{Sig}^{\mathbf{I}}(\cdot) := \langle \mathrm{Sig}(\cdot), \mathbf{I} \rangle$).

From here, the argument mirrors Cass & Turner (2024), which also underpins the alternative proof of Theorem 3.1. In order to apply Lemma 5, we would like to exchange the limit and expectation and the double sum. By Fubini-Tonelli and the dominated convergence theorem, to justify the exchange it is enough to show that

$$\sum_{|\mathbf{I}|, |\mathbf{J}|=0}^{\infty} \frac{1}{N^{\frac{|\mathbf{I} \star \mathbf{J}|}{2} + 1}} \mathbb{E}_{\xi_N} \left[ \left| \mathrm{tr} \left( A_{\mathbf{I} \star \mathbf{J}}^N \right) \right| \right] \left| \mathrm{Sig}^{\mathbf{I}}(\mathbb{X})_s \, \mathrm{Sig}^{\mathbf{J}}(\mathbb{Y})_t \right| \tag{25}$$

is uniformly bounded in $N$. As the matrices $A_i$ are Gaussian it holds that

$$\frac{1}{N^{\frac{|\mathbf{I} \star \mathbf{J}|}{2} + 1}} \mathbb{E}_{\xi_N} \left[ \left| \mathrm{tr} \left( A_{\mathbf{I} \star \mathbf{J}}^N \right) \right| \right] \leq \frac{1}{N^{\frac{|\mathbf{I} \star \mathbf{J}|}{2} + 1}} \mathbb{E}_{\xi_N} \left[ \left\| A_{\mathbf{I} \star \mathbf{J}}^N \right\|_{\mathrm{op}} \right] \leq \kappa^{|\mathbf{I} \star \mathbf{J}|} \Gamma \left( \frac{|\mathbf{I} \star \mathbf{J}|}{2} + 1 \right)$$

for a constant $\kappa$ and where $\| \cdot \|_{\mathrm{op}}$ is the operator norm.

Then, by the factorial decay of (the signature of) rough paths (Definition 2.1), and the fact that the $L_2$ norm is at least as large as the $L_1$ norm on $V^{\otimes n}$, there exists some $\omega > 0$ for which Eq. 25 is bounded by

$$\sum_{n,m=0}^{\infty} \frac{\Gamma \left( \frac{n+m}{2} + 1 \right) (\omega \kappa)^{m+n}}{\Gamma(\frac{n}{p} + 1) \Gamma(\frac{m}{q} + 1)} \leq \sum_{n,m=0}^{\infty} \frac{\sqrt{\Gamma(n+1) \Gamma(m+1)} (\omega \kappa)^{n+m}}{\Gamma(\frac{n}{p} + 1) \Gamma(\frac{m}{q} + 1)}$$

where the inequality follows from logarithmic convexity of $\Gamma$. By Stirling's formula,

$$\Gamma(\alpha n + 1) \asymp \sqrt{2\pi} \, (\alpha n)^{\alpha n + \frac{1}{2}} e^{-\alpha n} \quad \text{as } n \to \infty,$$

the numerator grows subfactorially relative to the product $\Gamma(n/p + 1)\Gamma(m/q + 1)$, so the double series converges for any fixed $(\omega, \kappa)$. Hence, by an exchange of limits and an application of Lemma 5, we see that

$$\lim_{N \to \infty} \frac{1}{N} \mathbb{E}_{\xi_N} \left[ \langle Z_s^N(\mathbb{X}), Z_t^N(\mathbb{Y}) \rangle_{\mathcal{H}} \right] = \sum_{|\mathbf{I}|, |\mathbf{J}|=0}^{\infty} \lim_{N \to \infty} \frac{1}{N^{\frac{|\mathbf{I} \star \mathbf{J}|}{2} + 1}} \mathbb{E}_{\xi_N} \left[ \mathrm{tr} \left( A_{\mathbf{I} \star \mathbf{J}}^N \right) \right] \mathrm{Sig}^{\mathbf{I}}(\mathbb{X})_s \, \mathrm{Sig}^{\mathbf{J}}(\mathbb{Y})_t$$

$$= \sum_{|\mathbf{I}|=0}^{\infty} \mathrm{Sig}^{\mathbf{I}}(\mathbb{X})_s \, \mathrm{Sig}^{\mathbf{J}}(\mathbb{Y})_t$$

$$= \langle \mathrm{Sig}(\mathbb{X})_s \, \mathrm{Sig}(\mathbb{Y})_t \rangle_{T((V))}$$

$$= K_{\mathrm{Sig}}^{\mathbb{X}, \mathbb{Y}}(s, t)$$

which concludes our proof. $\qquad \square$

# C ADDITIONAL EXPERIMENTAL RESULTS AND DETAILS

## C.1 EXPERIMENTAL SETUP

**Preprocessing.** We use the archive's pre-specified train/test splits. Each dataset is min–max scaled to $[-1, 1]$, and sequences are represented via piecewise-linear interpolation. We apply time and base-point augmentation, and tune the inclusion of a lead–lag transform via grid search. All sequences are resampled to length 200 following Toth et al. (2025).

**Implementation.** Random differential equation models are implemented in `Jax` in our library `RandomSigJax`. All the other benchmarks have been evaluated using the `KSig` library (Tóth et al., 2025) built on `CuPy` (Nishino & Loomis, 2017). Both libraries use `CuML` to perform SVM/LinearSVM calculations on GPU.

**Neural Baseline Configuration.** NCDE and NRDE are implemented with an MLP vector field and a linear readout. Their hidden dimensions are chosen so that the total parameter count approximately matches the feature budgets used for the random models, ensuring a fair comparison. Training follows the standard protocol: Adam optimiser, early stopping on validation accuracy.

**Compute.** All experiments were run on a single NVIDIA RTX 3090 GPU. Each approach is trained and evaluated 3 times on each dataset, then the median test accuracy is taken.

**Hyperparameter selection.** As documented in prior work on RFSF (Toth et al., 2025), truncated signature kernels (Király & Oberhauser, 2019), and SigPDE (Salvi et al., 2021a), the influence of the scaling parameters introduced in Eq. 10 and in Section 3.3 is strongly data- and task-dependent, and there is currently no principled procedure for selecting them a priori. Consistent with this literature, we treat them as hyperparameters tuned via grid search.

Some general heuristics nevertheless can guide the design of the search ranges: higher truncation orders $m$ are often beneficial for rougher paths (in the sense of larger $p$-variation), while reasonable choices of $\sigma_A, \sigma_B$ typically ensure that typical increments $\sigma_A \Delta X_t$ remain of order at most one.

**Hyperparameter grids.** We tune all models via grid search; the swept grids are listed below.

For all models using RFFs, the Fourier frequency scale is swept over $\{0.01, 0.025, 0.05, 0.1, 0.25, 0.5, 1, 2.5, 5, 10, 25, 50, 100\}\times$ the bandwidth suggested by Toth et al. (2025).

Model–specific ranges are:

- *RFSF*: signature level $M \in \{2, 3, 4, 5\}$.
- *All Random Differential Equation models*:
  - activation $\in \{\mathrm{id}, \tanh, \mathrm{ReLU}\}$;
  - $\sigma_A \in \{0.1, 0.25, 0.5, 0.75, 1.0, 1.25, 1.5, 2.0\}$;
  - $\sigma_B \in \{0.1, 0.25, 0.5\}$;
  - $\sigma_0 \in \{0.0, 0.5, 1.0, 1.5\}$.
- *R-RDE*: signature level $M \in \{2, 3, 4, 5\}$, *capped* so that the number of (log-)signature coordinates does not exceed the feature budget.
- *RF-CDE*: number of Fourier features $F \in \{32, 64, 128, 256, 512, 1024\}$, capped at $50\times$ the input dimension.

All feature-based models optionally apply feature normalisation (`True`/`False`).

Other baselines (SigPDE, RBF/GAK/RWS, SVMs) use standard grids over their key hyperparameters (kernel bandwidths, regularisation, warping parameters).

**Remark C.1.** *SigPDE is evaluated only with an RBF base kernel (Salvi et al., 2021a), due to the superior empirical performance reported therein.*

## C.2 Ablation: Number of Random Features

We study sensitivity to feature count by repeating the main evaluation with $N = 64$ and $N = 500$ random features. Tables 4 and 5 below mirror the main setting (training three runs per dataset and reporting mean accuracy).

| Dataset | R-CDE | RF-CDE | R-RDE | RFSF-DP | RFSF-TRP | NCDE | NRDE |
|---|---|---|---|---|---|---|---|
| ArticularyWordRecog. | 0.917 | **0.963** | 0.903 | 0.957 | 0.957 | 0.957 | 0.917 |
| AtrialFibrillation | 0.400 | 0.467 | **0.533** | 0.267 | 0.333 | 0.467 | 0.267 |
| BasicMotions | **1.000** | **1.000** | 0.975 | 0.975 | **1.000** | 0.975 | 0.975 |
| Cricket | 0.931 | **0.944** | 0.917 | 0.917 | **0.944** | 0.917 | 0.898 |
| EigenWorms | 0.420 | 0.611 | 0.594 | 0.701 | **0.755** | 0.675 | 0.420 |
| Epilepsy | **0.963** | **0.963** | 0.927 | 0.949 | 0.927 | **0.963** | 0.927 |
| EthanolConcentration | 0.308 | 0.338 | 0.361 | **0.430** | 0.418 | 0.385 | 0.361 |
| FingerMovements | 0.510 | **0.520** | 0.500 | 0.490 | 0.510 | 0.500 | 0.400 |
| Handwriting | 0.279 | **0.340** | 0.302 | 0.302 | 0.305 | 0.305 | 0.279 |
| Libras | 0.827 | **0.894** | 0.856 | 0.817 | 0.883 | 0.827 | **0.894** |
| NATOPS | **0.900** | **0.900** | 0.889 | 0.789 | 0.889 | 0.833 | 0.789 |
| RacketSports | 0.750 | **0.796** | 0.691 | 0.747 | 0.782 | **0.796** | 0.747 |
| SelfRegulationSCP1 | 0.825 | 0.853 | 0.857 | 0.880 | **0.884** | 0.846 | 0.857 |
| SelfRegulationSCP2 | 0.522 | 0.533 | **0.567** | 0.550 | 0.494 | 0.517 | 0.550 |
| StandWalkJump | 0.267 | **0.469** | 0.400 | 0.400 | 0.333 | 0.333 | 0.333 |
| UWaveGestureLibrary | 0.834 | 0.822 | **0.897** | 0.825 | 0.803 | 0.789 | 0.822 |
| Avg. acc. ($\uparrow$) | 0.666 | **0.713** | 0.698 | 0.687 | 0.701 | 0.692 | 0.652 |
| Avg. rank ($\downarrow$) | 4.437 | **2.500** | 4.094 | 4.156 | 3.375 | 4.186 | 5.250 |

Table 4: UEA test accuracies with $N = 64$ random features. The neural baselines NCDE and NRDE use comparable parameter budgets. Best result per row in **bold**.

| | R-CDE | RF-CDE | R-RDE | RFSF-DP | RFSF-TRP | SigPDE |
|---|---|---|---|---|---|---|
| ArticularyWordRecognition | 0.973 | **0.983** | 0.950 | 0.973 | **0.983** | **0.983** |
| AtrialFibrillation | 0.200 | 0.333 | **0.400** | **0.400** | 0.333 | 0.333 |
| BasicMotions | **1.000** | **1.000** | **1.000** | 0.975 | **1.000** | **1.000** |
| Cricket | **0.986** | **0.986** | 0.917 | 0.958 | 0.958 | 0.972 |
| EigenWorms | 0.458 | 0.664 | 0.612 | **0.824** | 0.817 | 0.794 |
| Epilepsy | 0.942 | **0.971** | 0.942 | 0.942 | 0.957 | 0.891 |
| EthanolConcentration | 0.319 | 0.407 | 0.375 | **0.517** | 0.414 | 0.460 |
| FingerMovements | 0.520 | **0.610** | 0.550 | 0.590 | 0.600 | **0.610** |
| Handwriting | 0.331 | 0.380 | 0.362 | 0.424 | **0.426** | 0.409 |
| Libras | 0.856 | **0.911** | 0.867 | 0.872 | 0.900 | 0.867 |
| NATOPS | 0.889 | **0.944** | 0.906 | 0.867 | 0.933 | 0.928 |
| RacketSports | 0.809 | 0.829 | 0.717 | **0.889** | 0.842 | 0.849 |
| SelfRegulationSCP1 | 0.877 | 0.881 | 0.843 | 0.894 | 0.881 | **0.904** |
| SelfRegulationSCP2 | **0.578** | **0.578** | 0.557 | 0.533 | 0.494 | 0.544 |
| StandWalkJump | 0.333 | 0.400 | **0.467** | 0.400 | 0.400 | 0.400 |
| UWaveGestureLibrary | 0.850 | 0.850 | **0.913** | 0.897 | 0.859 | 0.866 |
| Avg. acc. ($\uparrow$) | 0.683 | 0.733 | 0.711 | **0.747** | 0.737 | 0.738 |
| Avg. rank ($\downarrow$) | 4.812 | **2.812** | 4.125 | 3.187 | 3.031 | 3.031 |

Table 5: UEA test accuracies with $N = 500$ random features. SigPDE is a kernel method (no random features), so its results are unaffected by $N$. For each row, the best result is highlighted in **bold**.

**Performance at N=64 Random Features** Reducing the feature budget from 250 to 64 lowers accuracy across all models, but the relative ordering changes noticeably. RF–CDE achieves both the best average accuracy and the best average rank among all methods (0.713, 2.50), and R–RDE also remains competitive with only a modest drop (0.708 → 0.698). In contrast, the signature-projection baselines exhibit larger declines (RFSF-DP 0.726 → 0.687, RFSF-TRP 0.725 → 0.701). Easy datasets such as BasicMotions and Epilepsy stay saturated, while more complex datasets drive most of the differences across methods. Overall, the small-feature regime highlights that the random differential equation reservoirs retain good performance even when the feature budget is substantially constrained, whereas the neural baselines (NCDE, NRDE) struggle to remain competitive at this scale.

**Performance at N=500 Random Features.** The largest average gains appear for the randomized-signature baselines (RFSF-DP 0.726 → 0.747, RFSF-TRP 0.725 → 0.737), while RF-CDE remains essentially flat (0.741 → 0.733) and R-RDE shows a slight lift (0.708 → 0.711), with clearer improvements on harder sets (e.g., EigenWorms for RF-CDE: 0.630 → 0.664, UWaveGestureLibrary for R-RDE: 0.903 → 0.913). Easy tasks (e.g., BasicMotions) are already saturated at near-perfect accuracy. In short, our random differential equation reservoirs already operate efficiently at 250 features; doubling $N$ yields incremental benefits on a subset of challenging datasets.

## C.3 Additional Baselines for UEA Classification: Classical Kernels and RFF

For completeness, we report standard time-series baselines: Random Fourier Features (RFF), RBF kernel SVM, Global Alignment Kernel (GAK), and Random Warping Series (RWS). RFF is evaluated at two budgets (250 and 500 features), whereas RWS is reported only for the 250-feature setting. These baselines rarely achieve state-of-the-art performance on our suite but serve as useful reference points.

|  | RFF-250 | RFF-500 | RWS | GAK | RBF |
|---|---|---|---|---|---|
| ArticularyWordRecognition | 0.980 | 0.980 | 0.970 | 0.977 | 0.977 |
| AtrialFibrillation | 0.333 | 0.333 | 0.427 | 0.333 | 0.267 |
| BasicMotions | 0.925 | 0.925 | 0.995 | 1.000 | 0.975 |
| Cricket | 0.889 | 0.889 | 0.958 | 0.944 | 0.917 |
| EigenWorms | 0.431 | 0.431 | 0.578 | 0.511 | 0.496 |
| Epilepsy | 0.775 | 0.775 | 0.925 | 0.870 | 0.891 |
| EthanolConcentration | 0.316 | 0.316 | 0.284 | 0.361 | 0.346 |
| FingerMovements | 0.620 | 0.620 | 0.580 | 0.500 | 0.620 |
| Handwriting | 0.247 | 0.247 | 0.591 | 0.481 | 0.307 |
| Libras | 0.783 | 0.783 | 0.828 | 0.767 | 0.800 |
| NATOPS | 0.906 | 0.906 | 0.900 | 0.922 | 0.917 |
| RacketSports | 0.757 | 0.757 | 0.861 | 0.849 | 0.809 |
| SelfRegulationSCP1 | 0.894 | 0.894 | 0.829 | 0.915 | 0.898 |
| SelfRegulationSCP2 | 0.483 | 0.483 | 0.456 | 0.511 | 0.439 |
| StandWalkJump | 0.267 | 0.267 | 0.333 | 0.267 | 0.533 |
| UWaveGestureLibrary | 0.838 | 0.838 | 0.897 | 0.887 | 0.766 |
| Avg. acc. (↑) | 0.679 | 0.679 | 0.721 | 0.703 | 0.706 |

Table 6: Baseline comparison on UEA datasets: Random Fourier Features (250/500), Random Warping Series (RWS) – number of features = 250, Global Alignment Kernel (GAK), and RBF kernel SVM. Entries are test accuracies using the standard splits.

## C.4 Robustness to Missing Observations

We assess the stability of all models under partial observations using corrupted versions of the UEA multivariate time-series datasets. Importantly, the corruption mechanism is applied *only to the test set*, while training is always performed on the clean, unmodified training split. This isolates robustness at inference time.

This experiment uses a feature budget of $N = 64$, matching the configuration examined in the ablation study. Accordingly, the numbers reported here are directly comparable to the $N = 64$ results (Table 4) presented in Appendix C.2.

**Corruption model.**  Given an input path $X \in \mathbb{R}^{T \times d}$, we generate a binary mask $M \in \{0, 1\}^{T \times d}$ by independently removing entries with fixed probability $p$. We consider two regimes:

- **Low corruption:** $p = 20\%$,
- **High corruption:** $p = 40\%$.

The corrupted path is defined coordinatewise as

$$\widetilde{X}_{t,i} = \begin{cases} \text{missing}, & M_{t,i} = 0, \\ X_{t,i}, & M_{t,i} = 1. \end{cases}$$

Missing values are then imputed using simple linear interpolation along the time axis.

**Evaluation protocol.**  All models are trained on the full, uncorrupted training set. During testing, the corrupted–interpolated sequences are processed without modifying the training pipeline. Performance degradation as $p$ increases provides a direct measure of robustness to missing observations.

The hyperparameter grids and the general experimental setup follow exactly the Time-Series Classification experiment and are reported in Section C.1.

| | R-CDE | RF-CDE | R-RDE | RFSF-DP | RFSF-TRP |
|---|---|---|---|---|---|
| ArticularyWordRecog. | 0.896 | **0.953** | 0.903 | **0.953** | 0.943 |
| AtrialFibrillation | 0.400 | 0.400 | **0.533** | 0.333 | 0.200 |
| BasicMotions | 0.975 | **1.000** | 0.975 | 0.900 | **1.000** |
| Cricket | 0.944 | **0.972** | 0.875 | 0.875 | 0.931 |
| EigenWorms | 0.458 | 0.594 | 0.594 | 0.611 | **0.664** |
| Epilepsy | 0.906 | 0.913 | 0.768 | 0.913 | **0.984** |
| EthanolConcentration | 0.304 | 0.312 | 0.324 | **0.414** | 0.407 |
| FingerMovements | 0.510 | **0.520** | 0.510 | 0.510 | **0.520** |
| Handwriting | 0.279 | **0.302** | 0.279 | 0.300 | 0.228 |
| Libras | 0.867 | **0.906** | 0.733 | 0.806 | 0.878 |
| NATOPS | 0.844 | **0.867** | 0.844 | 0.778 | **0.867** |
| RacketSports | 0.743 | **0.763** | 0.539 | 0.612 | 0.697 |
| SelfRegulationSCP1 | 0.836 | **0.884** | 0.795 | 0.877 | 0.826 |
| SelfRegulationSCP2 | 0.533 | 0.505 | **0.550** | 0.483 | 0.483 |
| StandWalkJump | 0.267 | **0.400** | 0.333 | 0.267 | 0.200 |
| UWaveGestureLibrary | 0.817 | 0.817 | **0.857** | 0.806 | 0.819 |
| Avg. acc. ($\uparrow$) | 0.661 | **0.694** | 0.650 | 0.652 | 0.665 |
| Avg. rank ($\downarrow$) | 3.438 | **1.938** | 3.406 | 3.406 | 2.812 |
| Avg. acc. % decrease ($\downarrow$) | **0.695** | 2.672 | 6.777 | 5.074 | 5.082 |

Table 7: UEA test accuracies under **20**% missing observations, where the corruption is applied only to the test set. The number of features For each dataset, the best result is shown in **bold**.

**Results.**  Across the $20\%$ and $40\%$ missing-value settings, all models naturally experience a drop in accuracy, but the magnitude of this degradation varies substantially. *RF-CDE remains the strongest overall performer*, retaining the best average accuracy and best average rank in both corrupted regimes, with only moderate decreases ($-2.7\%$ at $20\%$ missing, $-5.7\%$ at $40\%$). R-CDE also degrades slowly ($-0.7\%$ and $-–5.9\%$), remaining competitive despite its simpler architecture. By contrast, the RFSF baselines exhibit noticeably larger losses, particularly at higher corruption levels.

Unexpectedly, R-RDE does not perform as well in this experiment, despite the theoretical robustness of signature features to missing data. Its accuracy drops more sharply than for CDE-based reservoirs. We suspect this is partly due to the short UEA time series and the relatively small chunk/step

size used in the log-ODE discretisation; in longer forecasting settings, or with larger log-signature chunks, we expect the intrinsic stability of signature features to manifest more clearly.

At the per-dataset level, performance declines are mostly monotone as corruption increases, with RF-CDE continuing to dominate on structured motion datasets, while R-RDE remains competitive on a few highly irregular tasks. Overall, the results suggest that CDE-based reservoirs are comparatively robust to missing inputs, while log-ODE RDEs may require longer temporal horizons or coarser signature blocks to fully leverage their theoretical advantages.

The results are reported in Tables 7 and 8.

|  | R-CDE | RF-CDE | R-RDE | RFSF-DP | RFSF-TRP |
|---|---|---|---|---|---|
| ArticularyWordRecog. | 0.840 | 0.940 | 0.840 | **0.957** | 0.937 |
| AtrialFibrillation | **0.400** | 0.333 | **0.400** | 0.333 | 0.200 |
| BasicMotions | 0.775 | **0.975** | 0.925 | 0.725 | 0.800 |
| Cricket | **0.972** | 0.944 | 0.847 | 0.819 | 0.931 |
| EigenWorms | 0.458 | 0.489 | 0.489 | 0.489 | **0.542** |
| Epilepsy | 0.884 | **0.913** | 0.645 | 0.826 | 0.760 |
| EthanolConcentration | 0.285 | 0.304 | 0.319 | **0.418** | 0.414 |
| FingerMovements | 0.510 | 0.520 | 0.500 | **0.560** | 0.540 |
| Handwriting | **0.279** | **0.279** | 0.261 | 0.267 | 0.221 |
| Libras | 0.830 | **0.883** | 0.733 | 0.783 | 0.850 |
| NATOPS | 0.710 | 0.801 | 0.755 | 0.755 | **0.844** |
| RacketSports | 0.651 | **0.697** | 0.454 | 0.539 | 0.579 |
| SelfRegulationSCP1 | 0.843 | **0.887** | 0.771 | 0.867 | 0.812 |
| SelfRegulationSCP2 | **0.522** | **0.522** | 0.517 | 0.517 | 0.478 |
| StandWalkJump | 0.267 | 0.469 | 0.400 | 0.333 | 0.267 |
| UWaveGestureLibrary | 0.803 | 0.812 | **0.821** | 0.812 | **0.821** |
| Avg. acc. ($\uparrow$) | 0.627 | **0.673** | 0.604 | 0.625 | 0.624 |
| Avg. rank ($\downarrow$) | 3.281 | **2.000** | 3.594 | 3.062 | 3.062 |
| Avg. acc. % decrease ($\downarrow$) | 5.858 | **5.651** | 13.36 | 9.058 | 10.89 |

Table 8: UEA test accuracies under **40**% missing observations, where the corruption is applied only to the test set. The number of features For each dataset, the best result is shown in **bold**.

## C.5 HURST CLASSIFICATION TASK

**Fractional Brownian Motion.** Fractional Brownian motion (fBm) $\{B_t^H\}_{t \in [0,1]}$ with Hurst exponent $H \in (0,1)$ is the unique mean-zero Gaussian process with covariance

$$\mathbb{E}\left[B_t^H B_s^H\right] = \frac{1}{2}\left(t^{2H} + s^{2H} - |t-s|^{2H}\right). \tag{26}$$

The parameter $H$ governs the *roughness* of sample paths:

- $H < \frac{1}{2}$ yields negatively correlated increments and rough trajectories;
- $H = \frac{1}{2}$ recovers standard Brownian motion;
- $H > \frac{1}{2}$ yields smoother trajectories with persistent increments.

For $H < \frac{1}{2}$, the expected $p$-variation is finite only for $p > 1/H$, so changes in $H$ induce clear geometric differences. Recovering $H$ from sample paths is therefore a natural benchmark for models based on signature features, kernels, or controlled/rough differential equations.

**Data Generation.** We generate univariate fractional Brownian motion using the standard Davies-Harte method. For each class
$$H \in \{0.05, 0.15, \ldots, 0.75\},$$
we produce:

- $n_{\text{train}} = 50$ samples per class,
- $n_{\text{test}} = 25$ samples per class,
- length $\ell = 256$ time steps,
- dimension $d = 3$ by stacking three independent fBm realisations with the same $H$.

Thus each input sample is a tensor $X \in \mathbb{R}^{N \times 3}$ and the label is the discrete index of the Hurst value.

**V1 and V2 Variants.** The first variant (**V1**) uses the raw fBm trajectories. The second variant (**V2**) applies a per-sample standardisation

$$\tilde{X}_{t,i} = \frac{X_{t,i} - \mu_i}{\sigma_i}, \qquad \mu_i = \frac{1}{N} \sum_{t=1}^{N} X_{t,i}, \qquad \sigma_i^2 = \frac{1}{N} \sum_{t=1}^{N} (X_{t,i} - \mu_i)^2. \tag{27}$$

This transformation removes global amplitude differences and forces the model to rely purely on geometric and temporal structure. Since fBm trajectories with different $H$ can have substantially different variances, this normalisation creates a more challenging benchmark in which roughness must be inferred independently of scale.

**Number of Features.** All Random Differential Equation models and RFSF baselines are evaluated with a feature budget of $N = 64$. The neural baselines (Neural CDE and Neural RDE) are configured to have a comparable number of parameters to ensure a fair comparison across architectures.

**Benchmarks.** In this experiment we compare our models exclusively with feature-based baselines: Random Fourier Signature Features (RFSF) with diagonal and tensor projections, Neural Controlled Differential Equations (NCDE) (Kidger et al., 2020), and Neural Rough Differential Equations (NRDE) (Morrill et al., 2021). We additionally evaluated plain Random Fourier Features, which consistently failed to exceed 30% accuracy in any configuration and are therefore omitted for clarity.

**Results.** The results are reported in Table 3 in Section 4.2.

