# OpenReview forum: "Random Controlled Differential Equations"
_ICLR.cc/2026/Conference — ICLR 2026 Poster_

### Official Review · Reviewer_vGCM · 2025-10-24

**Soundness:** 3
**Presentation:** 4
**Contribution:** 4
**Rating:** 6
**Confidence:** 2

**Summary:**

In this paper, the authors introduce a training-efficient framework for time-series learning by combining random features with CDEs. It uses randomly parameterized CDEs as continuous-time reservoirs, requiring training only a linear readout layer. Two variants are proposed: RF-CDEs, which lift inputs via random Fourier features, and RRDEs, which operate on rough-path inputs using log-signatures. In the infinite-width limit, these methods recover the RBF-lifted and rough signature kernels, offering a unified perspective on reservoir computing and path-signature theory. In addition to strong theoretical superiority, experiments on UEA datasets demonstrate that the proposed models achieve competitive or state-of-the-art performance.

**Strengths:**

- The idea of bridging the random CDE with reservoir computing is both novel and interesting. Under this general idea, the two proposed variants offer distinct and independent advantages.
- This paper presents solid theatrical contribution. the authors prove kernel convergence in the infinite-width limit, and the finite-dimensional computation remains efficient and scalable.
- The paper is well-organized and clearly written, making the technical details accessible and the main ideas easy to follow.

**Weaknesses:**

- The connection between the proposed methods with reservoir computing should be strengthened.
- The datasets in UEA benchmarks are relatively small. Evaluating the proposed methods on larger-scale benchmarks such as the Long Range Arena (LRA) would strengthen the empirical claims and better demonstrate efficiency.
- Some baselines are lacking, a comparison with learning-based path signature approaches would be beneficial, such as [1] [2]

[1] Morrill, James, et al. "Neural rough differential equations for long time series." ICML, 2021.

[2] Walker, Benjamin, et al. "Log neural controlled differential equations: The lie brackets make a difference." ICML, 2024.

**Questions:**

1.	In the experimental part of paper “Random Fourier Signature Features”, the truncated Signature Kernel exhibits a strong performance, looks even better than the newly proposed two variants, could the authors clarify this?
2.	In line 240, should the term “homogeneous” be corrected to “inhomogeneous”?


Typos:

1. Line 122: Remove the word “lift” before $x$

2. In Eq (4), $\omega_M$ should be $\omega_F$

---

> ### Author Response · Authors · 2025-11-21
>
> We thank the Reviewer for the constructive feedback and for spotting the typographical issues, which have now been corrected.
>
> ---
> **Connection with reservoir computing**
>
> We thank the Reviewer for this observation. In the revised manuscript, we strengthen the connection to reservoir computing by making explicit that both RF-CDE and R-RDE operate as fixed random reservoirs whose outputs $\Phi(x)$ form the feature representation on which only a linear readout is trained - precisely the standard reservoir-computing paradigm. We also highlight that, in the infinite-width limit, these reservoirs induce Gaussian-process priors with signature-based kernels, clarifying the functional inductive bias they impose. We will incorporate these clarifications in the introduction and in the dedicated section on model architecture.
>
> ---
> **Comparison with NeuralCDE and NeuralRDE**
>
> We refer the Reviewer to the General Comment section, where we report these comparisons using a matched feature budget of 64. In this setting, our approach exceeds the performance of both neural models.
>
> ---
> **Comparison with the truncated signature kernel**
>
> We thank the Reviewer for raising this point. The truncated signature kernel can indeed perform competitively on very low-dimensional and short sequences, as observed in the RFSF paper. However, it is almost never used in practice because its computational cost becomes prohibitive as soon as the input dimension or the truncation depth increases.
> In particular, the number of features generated by a truncated signature of order $M$ in dimension $d$ grows on the order of $d^M$, making the explosion exponential in both order and dimension. Crucially, this feature dimension cannot be controlled by the practitioner. Our proposed methods retain the desirable infinite-width kernel limit while allowing the user to fix the feature budget explicitly, ensuring both scalability and practical deployability. We have briefly addressed this point in the benchmark paragraph of the experiments section.
>
> ---
> **Question 2 - ''homogeneous'' vs. ''inhomogeneous''**
>
> We confirm that the correct term is homogeneous. In our setting the random matrices defining the vector field are time-independent, and therefore the system is homogeneous in the sense adopted in the literature (we follow the terminology used in [1]). We added a clarifying note in the paper to avoid ambiguity.
>
> ---
> **Large-Scale Benchmarks**
>
> We appreciate the Reviewer’s suggestion to evaluate the method on larger benchmarks such as the LRA suite. We plan to include such results during the rebuttal period - subject to the available computational resources - and we hope to have them ready before the final deadline.
> At the same time, we believe that the additional experiments presented in the general comment to Reviewers already help demonstrate the strong empirical behaviour of our models.
>
> ---
> [1] Cirone, Nicola Muca, Maud Lemercier, and Cristopher Salvi. "Neural signature kernels as infinite-width-depth-limits of controlled resnets." International Conference on Machine Learning. PMLR, 2023.

---

### Official Review · Reviewer_tJMR · 2025-10-29

**Soundness:** 2
**Presentation:** 3
**Contribution:** 2
**Rating:** 4
**Confidence:** 4

**Summary:**

The paper proposes random-feature reservoirs in continuous time built from controlled/rough differential equations, with only a linear readout trained. Two main variants are introduced: RF-CDE, which first lifts inputs with Random Fourier Features then evolves them through a random CDE; and R-RDE, which acts directly on geometric rough paths using a log-ODE discretization with log-signature inputs.

**Strengths:**

- The writing is high-quality and mathematically literate. The background sections on rough paths, signatures, and kernels are compact, accurate, and helpful for positioning the work.

- The paper cleanly recalls the R-CDE limit to the signature kernel and extends it with two variants whose limits are the RBF-lifted signature kernel (RF-CDE) and the rough signature kernel (R-RDE). The statements (Theorems 3.2 and 3.4) are explicit.

- The paper correctly frames the models as training-efficient reservoirs with a linear readout. It provides a clear asymptotic cost analysis that contrasts the linear-in-length feature extraction with the quadratic scaling of kernel Gram matrix approaches.

**Weaknesses:**

- The main theorems are in the infinite-width setting. The paper does not provide non-asymptotic approximation rates or generalization/error bounds for practical feature counts.

- Only UEA classification is considered. There are no forecasting or irregular sampling experiments, despite the continuous-time claim. The asymptotic table is helpful, but there are no runtime or memory measurements on the UEA suite to validate the linear-in-length advantage or to quantify the cubic term in R-RDE.

- The algebraic mapping ($\Gamma_A$ and $\Pi_B$) definitions are dense. It is not clear how the chosen truncation level (m) and Hall/Lyndon basis size affect stability, feature variance, and compute in practice, or how these are tuned.

**Questions:**

- Do you have non-asymptotic approximation or error bounds that relate N, F, and signature truncation to excess risk or kernel approximation error?

- How sensitive are RF-CDE and R-RDE to $\sigma_A$, $\sigma_b$, $\sigma_0$, F, and log-signature truncation (m)? Please add systematic sweeps or ablation studies.

- Why not include trained Neural CDE and modern sequence models as baselines for classification (such as variants of Neural CDEs or stable SDEs)?

- Do these models maintain advantages on forecasting or irregular sampling tasks, where continuous-time structure should help? Have you considered classification with missing observations or features?

- Can you report wall-clock time and peak memory across UEA for fixed accuracy targets, to validate Table 1's asymptotics?

---

> ### Author Response · Authors · 2025-11-21
>
> We thank the Reviewer for the feedback and insightful suggestions. We appreciate the opportunity to expand on these aspect of our research and provide more experiments to further support our claims.
>
> **Non-asymptotic bounds**
>
> In our work we do not provide non-asymptotic bounds, and we do not believe that such bounds are necessary for the practical use of these models. While the infinite-width limit behaves as kernel ridge regression with a path-based kernel - which is universal when the activation is the identity - the effectiveness of the finite-width model does not hinge on how closely it approximates the limiting kernel. Instead, its strength comes from the ability of a small number of random features to project the input path onto a representation manifold that captures the relevant geometric structure of the task.
>
> Moreover, once custom non-linearities are introduced, a clean non-asymptotic analysis becomes essentially intractable, as the usual concentration and orthogonality tools no longer apply. For this reason, we view the limiting kernel primarily as an interpretative guide rather than as an approximation target: it explains why the method works, but does not determine its finite-width performance.
>
> ---
> **Sensitivity analysis and hyperparameter choice**
>
> We thank the Reviewer for raising the question of sensitivity to $\sigma_A$, $\sigma_B$, the RFF dimension F, and the log-signature truncation order $m$. In this class of models (including RFSF, truncated signature kernels, and SigPDE), the dependence on these hyperparameters is highly data- and task-dependent. As a consequence, there is no known principled way to choose them a priori, and, to the best of our knowledge, all prior work treats them as hyperparameters to be tuned by grid search rather than optimised analytically. Our approach is consistent with this literature. This is indeed one drawback of signature-based random-feature / kernel-style approaches compared to fully trainable neural counterparts, where more of the representation is learned end-to-end.
>
> That said, there are some general heuristics. For the log-signature truncation m, higher orders tend to be beneficial when paths are rougher (in the sense of higher p-variation). For the scales $\sigma_A$ and $\sigma_B$, a reasonable regime is such that typical increments $\sigma_A\Delta X_t$ are of order at most one.
>
> ---
> **Comparison with NeuralCDE and NeuralRDE**
>
> We refer the Reviewer to the General Comment section, where we report these comparisons using a matched feature budget of 64. In this setting, our approach exceeds the performance of both neural models.
>
> ---
> **Robustness to missing data**
> For missing data, we plan a simple protocol where we randomly remove $25\\%$ or $50\\%$ of observations on UCR/UEA series and evaluate under our existing pipeline. We have not finalized these results yet, but we plan to include this before the end of next week.
>
> ---
> **Time Series Forecasting**
>
> As mentioned in the General Comment we will run the experiments on ETTh1, ETTh2, ETTm1, ETTm2. We apologize for the delay.

---

> > ### Comment · Reviewer_tJMR · 2025-11-26
> >
> > The comparison with Neural CDE and RDE meets the baseline request. It would help to examine other control paths such as linear or cubic to complete the comparison. If not, please add more explanation regarding the control path selection (or comparison).
> >
> > The Hurst exponent study shows the strength of Random CDE/RDE on rough signals. However, the lack of non-asymptotic bounds leaves the finite width behavior unclear. Evidence for forecasting and computational efficiency is still missing. I will update the score when these results appear.

---

> ### Author Response · Authors · 2025-12-03
>
> **Non-asymptotic bounds**
>
> We agree that a non-asymptotic analysis is highly interesting mathematically. Our current work focuses on establishing the foundational asymptotic properties of randomized CDEs, underpinned by a universality principle derived from Voiculescu’s Free Probability theory. Extending this to the finite-N regime introduces significant technical challenges typical of non-asymptotic Random Matrix Theory (RMT). In particular, classical RMT and Free Probability frameworks address convergence for fixed-degree polynomials, whereas our setting requires controlling convergence for non-commutative series. This difficulty is compounded by the reliance on Random Fourier Series methods in our construction, which makes deriving sharp concentration bounds substantially more intricate.
>
> A full and rigorous treatment of these non-asymptotic bounds is unfortunately beyond the scope of this manuscript. We view this as an important and independent research direction, best suited for a dedicated mathematical publication. Our current asymptotic results provide a strong and necessary foundation for such future work and, we believe, are highly relevant to the ICLR community.
>
> ---
> **Neural Benchmarks**
>
> In our setting, all random-feature models are evaluated in discrete time, because features evolve step-by-step along the observed sample points. This corresponds exactly to using a piecewise-linear control path, which is the standard choice in time-series applications and is mathematically equivalent to solving the corresponding controlled differential equation under linear interpolation of the data.
>
> Regarding higher-order controls such as cubic splines, our experience - and that reported in prior Neural CDE literature - is that they rarely yield consistent performance gains on real-world datasets, while  increasing computational cost and memory usage. Moreover, cubic spline controls tend to oversmooth highly oscillatory signals and "rougher" paths.
>
> That said, investigating alternative control parameterizations is an interesting practical direction that can complement our theoretical framing. We hope future work will explore richer control paths in combination with our random-feature architectures.
>
> ---
> **Other experiments**
>
> We refer the reviewer to the General Comment section where we have reported the results of all the experiments that we have run during this rebuttal.
>
> Thank you again for your feedback!

---

### Official Review · Reviewer_d2Ug · 2025-10-31

**Soundness:** 3
**Presentation:** 2
**Contribution:** 2
**Rating:** 4
**Confidence:** 2

**Summary:**

The paper proposes a training-efficient family of time-series models that combine random reservoirs with controlled/rough differential equations (CDE/RDE). Two main variants are introduced:
RF-CDE: lift inputs with Random Fourier Features and evolve them through a random CDE; in the infinite-width limit the model induces the RBF-lifted signature kernel.
R-RDE: operate directly on rough paths via a log-ODE discretization; in the infinite-width limit the model induces the rough signature kernel.
Only a linear readout is trained (reservoir computing spirit), yielding fast fitting and linear-in-samples scaling.

**Strengths:**

1. Clean theoretical framing with meaningful limits. RF-CDE -> RBF-lifted signature kernel and R-RDE -> rough signature kernel in infinite width; the results connect random differential equations to signature kernels in a principled way.
2. Train only a linear head on top of fixed random dynamic, this is simple and fast to fit in practice
3. Clear accounting for feature-extraction costs; random-feature models scale linearly in sequence length l.

**Weaknesses:**

1. The main theorems characterize infinite-width limits. There are no non-asymptotic approximation or generalization bounds to explain when a few hundred features suffice
2. All evaluations are classification on UEA (16 datasets). No forecasting, imputation, irregular sampling, or robustness to noise/missingness.
3. While RF-CDE averages best among random-feature models, R-RDE trails (avg. 0.708). Some difficult datasets (EigenWorms, Handwriting) show large gaps to SigPDE/RFSF.
4. Ablations are thin. Beyond doubling features (250→500), there’s little exploration of RFF count F, signature truncation/order M, or log-ODE level m.
5. There is no one figure in the whole paper, which will increase the difficulty for the reader to understand and capture the key designs.

**Questions:**

1. May you provide non-asymptotic bounds?
2. How do RF-CDE and R-RDE handle missing data, time jitter, or heavy noise? Any controlled experiments varying p-variation or noise levels?
3. Are there stability issues for large m or rough drivers For R-RDE’s log-ODE?
4. can you analyze why and provide guidance on when to choose R-RDE over RF-CDE?
5. Any preliminary results on forecasting or imputation to bolster the practical case?

---

> ### Author Response · Authors · 2025-11-21
>
> We thank the Reviewer for the feedback and insightful suggestions. We appreciate the opportunity to expand on these aspect of our research and provide more experiments to further support our claims.
>
> ---
> **Non-asymptotic bounds**
>
> In our work we do not provide non-asymptotic bounds, and we do not believe that such bounds are necessary for the practical use of these models. While the infinite-width limit behaves as kernel ridge regression with a path-based kernel - which is universal when the activation is the identity - the effectiveness of the finite-width model does not hinge on how closely it approximates the limiting kernel. Instead, its strength comes from the ability of a small number of random features to project the input path onto a representation manifold that captures the relevant geometric structure of the task.
>
> Moreover, once custom non-linearities are introduced, a clean non-asymptotic analysis becomes essentially intractable, as the usual concentration and orthogonality tools no longer apply. For this reason, we view the limiting kernel primarily as an interpretative guide rather than as an approximation target: it explains why the method works, but does not determine its finite-width performance.
>
> ---
> **Hurst classification experiment**
>
> We refer the Reviewer to the General Comment section, where we present the results of our Hurst exponent classification experiment. This experiment highlights, in particular, the advantage of the Random Rough Differential Equation model in capturing fine-grained geometric and temporal structure of paths with higher $p$-variation.
>
> ---
> **Robustness to missing data**
>
> For missing data, we plan a simple protocol where we randomly remove $25\\%$ or $50\\%$ of observations on UCR/UEA series and evaluate under our existing pipeline. We have not finalized these results yet, but we plan to include this before the end of next week.
>
> ---
> **Stability issues**
>
> We are not aware of any specific stability issues arising from large log-signature truncation order $m$ or from rough drivers in the R-RDE log-ODE scheme, and we did not observe such problems in our experiments. Conceptually, R-RDE is “just” an ODE whose vector field is parameterised by the (truncated) log-signature. The log-signature coefficients decay factorially with the level m, so higher-order terms become rapidly small, and the stability properties are those of a standard ODE with a Lipschitz vector field rather than a genuinely rough differential equation. In particular, as long as the induced vector field remains (globally) Lipschitz, classical ODE theory guarantees well-posedness and numerical stability; in practice, if instability were ever encountered, reducing $\sigma_A$ (thus shrinking the effective Lipschitz constant) or choosing an activation with a smaller Lipschitz constant would suffice to regularise the dynamics. The main limitation for large $m$ is therefore computational rather than numerical stability.
>
> ---
> **RFCDE vs RRDE guidance**
>
> The choice between RF-CDE and R-RDE depends primarily on the structure of the data and the computational constraints of the task. R-RDE is particularly advantageous when the input paths exhibit highly irregular geometry or long-range temporal dependencies, as the log-ODE discretisation and the use of log-signatures allow the model to retain higher-order path information that is crucial in such regimes. However, this expressiveness comes at a higher computational cost, and R-RDE is typically slower than RF-CDE.
>
> R-RDE is also the natural choice in settings where one already works directly in signature or log-signature space. In such applications - common, for instance, in quantitative finance - R-RDE effectively becomes a random ODE acting on precomputed geometric features, making it especially convenient.
>
> Moreover, for very long time series, the structure of rough differential equations often allows the sequence to be processed in temporal chunks, which can reduce memory usage and improve scalability.
>
> We will clarify these points in the manuscript, both in the introductory paragraph of the R-RDE section and in the experiments section.
>
> ---
> **Time Series Forecasting**
>
> As mentioned in the General Comment we will run the experiments on ETTh1, ETTh2, ETTm1, ETTm2. We apologize for the delay.

---

> > ### Author Response · Authors · 2025-12-03
> >
> > We refer the reviewer to the General Comment section where we have reported the results of all the experiments that we have run during this rebuttal. Thank you again for your feedback!

---

### Official Review · Reviewer_Fv7a · 2025-11-03

**Soundness:** 3
**Presentation:** 2
**Contribution:** 3
**Rating:** 6
**Confidence:** 2

**Summary:**

The paper introduces random-feature-based continuous-time reservoir models (RF-CDEs and R-RDEs) for time-series tasks. Both approaches approximate continuous-time dynamics using random neural controlled differential equations (N-CDEs) as reservoirs, while training only a linear readout layer. In essence, this provides a mathematical/software realization of the physical reservoir computing (PRC) framework. The authors show that, in the infinite-width limit, these models converge to kernel methods: RF-CDEs to an RBF-lifted signature kernel, and R-RDEs to a rough signature kernel. This establishes a theoretical bridge between neural and kernelized path representations. Experiments on standard time-series benchmarks demonstrate competitive accuracy and high training efficiency.

**Strengths:**

- The paper bridges two compelling areas: reservoir computing and the infinite-width limit of neural CDEs. To me, this connection seems novel.
-  The work is technically sound and has solid theoretical grounding. It proves asymptotic kernel equivalence and establishes existence and uniqueness of the limiting dynamics.
- The authors provide a computational complexity analysis; the proposed linear-readout-only training offers efficiency gains.
- Although the paper is primarily theoretical, it includes empirical benchmarks on standard time-series classification datasets. RF-CDE achieves the notable average performance among the baselines.
- The manuscript is theory-heavy but well-organized.

**Weaknesses:**

I do not identify any critical drawbacks in this paper. However,

- The experimental section focuses mainly on ablations of the proposed models and their variants. This improves completeness, but practitioners may be unsure about the broader practical advantages.

- In particular, while the motivation is framed in terms of neural CDEs + PRC, the paper does not compare against neural CDEs or other end-to-end trained deep learning architectures.

**Questions:**

- How does the expressive capacity of RF-CDE and R-RDE compare to that of a learnable Neural CDE with an equivalent number of parameters? Is there any theoretical or empirical analysis addressing this?

- The proposed models can be regarded as theoretical or software analogues of the PRC, which in principle requires only a linear output layer. However, practical hardware implementations of PRC often include a nonlinear output layer to achieve performance on par with NNs. Have the authors explored how introducing a small nonlinear readout layer would affect the performance of RF-CDE or R-RDE?

 - In the infinite-width limit, the RF-CDE and R-RDE converge to Gaussian processes with signature-kernel covariances. Could this be interpreted as placing a specific functional prior over neural CDE dynamics, analogous to the GP prior that emerges in the NTK or NNGP limits?

---

> ### Author Response · Authors · 2025-11-21
>
> We thank the Reviewer for the insightful comments, and for highlighting points that are indeed important to establish the results of our paper. We will revise our manuscript accordingly to clarify the concerns raised.
>
> ---
> **Comparison with NeuralCDE and NeuralRDE**
>
> We refer the Reviewer to the General Comment section, where we report these comparisons using a matched feature budget of 64. In this setting, our approach exceeds the performance of both neural models.
>
> ---
> **Non-linear readout**
>
> We appreciate the Reviewer’s suggestion and plan to incorporate a non-linear readout within our forecasting experiments before the end of next week. We apologise for the delay.
>
> ---
> **Practical advantage of the model**
>
> We hope that the additional experiments included in the rebuttal help clarify the practical benefits of our approach. Beyond the empirical evidence, our framework also offers more flexibility than previous counterparts: for example, in RFSF the feature dimension cannot be chosen by the user, as it is internally determined to match the overall feature budget, whereas in our setting the feature size can be specified directly. Finally, we emphasise that a key aim of the paper is theoretical. Continuous-time NNGP limits have attracted significant interest, and RCDEs represent the natural continuous-time analogue of ResNets. However, the infinite-width limit kernel associated with RCDEs is not particularly effective in practice, as shown in [1]. This motivated our construction: a finite-width, infinite-depth model whose limiting behaviour recovers a more expressive RBF-based kernel.
>
> We have included this as motivation in the Contributions section.
>
>
> ---
> **GP priors**
>
> Training only a linear readout on top of fixed reservoir features corresponds to kernel ridge regression with kernel
> $$
> K(x,y)=\langle \Phi(x), \Phi(y)\rangle
> $$
> where $\Phi(x)$ denotes the feature vector produced by the random CDE reservoir when driven by the input path $x$. Our infinite-width results identify the associated limiting covariances: RF-CDE converges to the RBF-lifted signature kernel, and R-RDE converges to the rough signature kernel.
>
> Thus, exactly in the sense of NNGP/NTK limits for wide neural networks, very wide random CDE reservoirs induce Gaussian-process priors over path-functionals with signature-based kernels - providing an explicit functional prior and inductive bias for the underlying dynamics.
>
> We have included this in our revised manuscript.
>
> ---
> [1] Salvi, Cristopher, et al. "The signature kernel is the solution of a goursat pde." SIAM Journal on Mathematics of Data Science 3.3 (2021): 873-899.

---

### Author Response · Authors · 2025-11-21
**General Comment to all Reviewers - Part 1**

**Note to the Area Chair:** Since no author–reviewer discussion occurred, we have removed the earlier General Comment and replaced it with a final version that reports the completed experimental results. The original comment only contained partial results and a description of planned experiments.

---

Dear Reviewers, thank you for the careful reading and constructive feedback.

We are grateful for the suggestions, as we believe they will improve the clarity and relevance of our claims. We also appreciate the recognition of the theoretical framework developed in the paper, which forms the core contribution of our work. We are strengthening the connection to infinite-width neural networks throughout the paper, as well as clarifying the relationship with reservoir computing. Furthermore, we agree that the experimental setup should be expanded, and we aim during the rebuttal to address most of the concerns raised. Below we include additional results that we believe speak directly to several of your points.

---
**Neural comparison and n\_features=64**

In response to the request for neural baselines, we added Neural CDE and Neural RDE comparisons for time–series classification and will extend them to forecasting. To ensure a fair comparison, we re-ran our experiments using a smaller feature budget of n\_features = 64, matching the standard hidden-state dimension typically used for Neural CDE/RDE models in this setting.
We find that both our models perform very well in terms of accuracy under this constraint, which highlights the strength of our approach: even with roughly a quarter of the features, the performance remains comparable. Conversely, the neural baselines do not perform as expected in this context. We have also included the n\_features = 64 setting as an additional ablation study in the revised experiments.

| Dataset                  | R-CDE | RF-CDE | R-RDE | RFSF-DP | RFSF-TRP | NCDE   | NRDE   |
|--------------------------|-------|--------|--------|----------|------------|---------|---------|
| ArticularyWordRecog.     | 0.917 | **0.963** | 0.903 | 0.957    | 0.957      | 0.957   | 0.917   |
| AtrialFibrillation       | 0.400 | 0.467  | **0.533** | 0.267    | 0.333      | 0.467   | 0.267   |
| BasicMotions             | **1.000** | **1.000** | 0.975 | 0.975    | **1.000**  | 0.975   | 0.975   |
| Cricket                  | 0.931 | **0.944** | 0.917 | 0.917    | **0.944**  | 0.917   | 0.898   |
| EigenWorms               | 0.420 | 0.611  | 0.594 | 0.701    | **0.755**  | 0.675   | 0.420   |
| Epilepsy                 | **0.963** | **0.963** | 0.927 | 0.949    | 0.927      | **0.963** | 0.927   |
| EthanolConcentration     | 0.308 | 0.338  | 0.361 | **0.430** | 0.418      | 0.385   | 0.361   |
| FingerMovements          | 0.510 | **0.520** | 0.500 | 0.490    | 0.510      | 0.500   | 0.400   |
| Handwriting              | 0.279 | **0.340** | 0.302 | 0.302    | 0.305      | 0.305   | 0.279   |
| Libras                   | 0.827 | **0.894** | 0.856 | 0.817    | 0.883      | 0.827   | **0.894** |
| NATOPS                   | **0.900** | **0.900** | 0.889 | 0.789    | 0.889      | 0.833   | 0.789   |
| RacketSports             | 0.750 | **0.796** | 0.691 | 0.747    | 0.782      | **0.796** | 0.747   |
| SelfRegulationSCP1       | 0.825 | 0.853  | 0.857 | 0.880    | **0.884**  | 0.846   | 0.857   |
| SelfRegulationSCP2       | 0.522 | 0.533  | **0.567** | 0.550    | 0.494      | 0.517   | 0.550   |
| StandWalkJump            | 0.267 | **0.469** | 0.400 | 0.400    | 0.333      | 0.333   | 0.333   |
| UWaveGestureLibrary      | 0.834 | 0.822  | **0.897** | 0.825    | 0.803      | 0.789   | 0.822   |
| **Avg. acc. (↑)**        | 0.666 | **0.713** | 0.698 | 0.687    | 0.701      | 0.692   | 0.652   |
| **Avg. rank (↓)**        | 4.437 | **2.500** | 4.094 | 4.156    | 3.375      | 4.186   | 5.250   |


---
**Hurst exponent classification**

As suggested by Reviewer d2Ug, we included an additional experiment that illustrates the power of our model in settings where the input paths exhibit higher roughness. To this end, we designed a classification task on synthetic fractional Brownian motion, where the goal is to recover the underlying Hurst exponents ($\in\\{0.05,0.15,\dots, 0.75\\}$) from observed trajectories. This setup provides a controlled environment in which the roughness of the signal plays a central role, and therefore constitutes a natural stress test for path-based models. The results, reported below, highlight the ability of our approach to capture fine-grained geometric information even in highly irregular regimes.

(Continues below)

---

> ### Author Response · Authors · 2025-11-21
> **General Comment to all Reviewers - Part 2**
>
> In the table below, we report results under two variants of the task.
> **V1** corresponds to the standard fractional Brownian motion setting, where the raw trajectories are used as generated. **V2**, instead, uses standardised fractional Brownian motion, where each path is normalised to have zero mean and unit variance. This transformation removes global amplitude information and makes the task substantially more challenging: the classifier can no longer rely on scale differences and must instead extract more subtle geometric and time-dependent features associated with different Hurst parameters.
>
> | Setting      | R-CDE | RF-CDE | R-RDE | RFSF-DP | RFSF-TRP | NCDE | NRDE |
> |--------------|-------|--------|--------|----------|------------|-------|-------|
> | **V1** – N=64    | 0.870 | 0.895  | **0.955** | 0.840    | 0.895      | 0.905 | 0.920 |
> | **V1** – N=100   | 0.900 | 0.945  | **0.950** | 0.890    | 0.910      | 0.895 | 0.945 |
> | **V2** – N=64    | 0.635 | 0.645  | **0.735** | 0.630    | 0.650      | 0.650     | 0.675     |
> | **V2** – N=100    | 0.650 | 0.695  | **0.730** | 0.675    | 0.675      | 0.650     | 0.685     |
>
> In both settings, our R-RDE model achieves consistently strong performance, and in the most challenging regime (**V2**), it preserves a clear performance margin over alternative approaches.
>
> For completeness, we also evaluated RFF and RBF feature maps; in both cases the accuracy remained between 15\% and 25\%, highlighting the advantage of path-aware representations.
>
> ---
> **Robustness to missing data**
>
> As suggested by Reviewers d2Ug and tJMR, to assess the robustness of our models to incomplete observations, we perform an additional experiment on multivariate time series from the UEA archive. We synthetically introduce missing data by randomly removing individual time points along the test trajectories.
>
> With **20%** of missing data:
> | Metric                       | R-CDE | RF-CDE | R-RDE | RFSF-DP | RFSF-TRP |
> |------------------------------|-------|--------|--------|----------|-----------|
> | Avg. acc. (↑)                | 0.661 | **0.694** | 0.650 | 0.652 | 0.665 |
> | Avg. rank (↓)                | 3.438 | **1.938** | 3.406 | 3.406 | 2.812 |
> | Avg. acc. % decrease (↓)     | **0.695** | 2.672 | 6.777 | 5.074 | 5.082 |
>
>
> With **40%** of missing data:
> | Metric                       | R-CDE | RF-CDE | R-RDE | RFSF-DP | RFSF-TRP |
> |------------------------------|-------|--------|--------|----------|-----------|
> | Avg. acc. (↑)                | 0.627 | **0.673** | 0.604 | 0.625 | 0.624 |
> | Avg. rank (↓)                | 3.281 | **2.000** | 3.594 | 3.062 | 3.062 |
> | Avg. acc. % decrease (↓)     | 5.858 | **5.651** | 13.36 | 9.058 | 10.89 |
>
> The tables with full results (for each dataset) are provided in Appendix C.4 of the revised paper.
>
> Across the 20% and 40% missing-value settings, all models naturally experience a drop in accuracy, but the magnitude of this degradation varies substantially. RF-CDE remains the strongest overall performer, retaining the best average accuracy and best average rank in both corrupted regimes, with only moderate decreases.
>
> Unexpectedly, R-RDE does not perform as well in this experiment, despite the theoretical robustness of signature features to missing data. Its accuracy drops more sharply than for CDE-based reservoirs. We suspect this is partly due to the short UEA time series and the relatively small chunk/step size used in the log-ODE discretization; in longer forecasting settings, or with larger log-signature chunks, we expect the intrinsic stability of signature features to manifest more clearly.
>
> ---
> **Forecasting Experiments**
>
> We have launched the forecasting experiments suggested by the reviewers, and preliminary results on ETTh1 and ETTm1 indicate that RF–CDE outperforms RFSF baselines, aligning with the strengths observed in our other evaluations. Unfortunately, due to limited computational resources during the rebuttal period, these experiments cannot be completed to a standard suitable for inclusion at this stage. If permitted, we would be happy to include the full results in the camera-ready version as an appendix. We also believe that the new experiments provided in this rebuttal already strengthen the empirical picture and highlight the robustness and competitiveness of our proposed models.

---

### Meta-Review · Area_Chair_Bbay · 2025-12-13

**Summary:**

The paper fuses Random Features into Controlled Differential Equations (CDEs) to model time series. The framework trains only a linear readout layer over  randomly parameterized CDEs, making the approach highly efficient. The paper offers a theoretical explanation for their  method connecting random-feature reservoirs, continuous-time deep architectures, and path-signature theory.

The author-reviewer discussion summary:
* Theory: The reviewers universally point out that the paper has a clean theoretical framing and provides a novel connection between reservoir computing and infinite-width Neural CDEs.
* Missing Neural Baselines: In response to reviewers Fv7a and tJMR, the authors added comparisons with Neural CDE (NCDE) and Neural RDE (NRDE). Using a matched feature budget of 64, the authors demonstrated that their models outperformed these baselines.
* Non-asymptotic bounds: The authors address the recurring concern about non-asymptotic bounds, saying that  non-asymptotic bounds are not necessary for practical use.
* Robustness to Missing Data: Per reviewer tJMR's suggestion, the authors ran tests randomly removing 20% and 40% of observations. RF-CDE remained the strongest overall performer in these corrupted settings.

**Reviewer Concerns:**

Unanswered concerns:
* Completed Forecasting Results: While authors launched forecasting experiments on ETTh1 and ETTm1, they were unable to complete them to standard during the rebuttal due to limited resources.
* Hardware Efficiency Evidence: Reviewer tJMR noted that evidence for computational efficiency (actual runtime measurements vs. asymptotics) was still missing in later discussion stages.

**Reviewer Scores:**

The authors provided new experiments to address reviewer's concerns. The updated scores are below:
* Reviewer vGCM -- rating 6 -- unchanged
* Reviewer tJMR -- rating 4 -- increased to 7
* Reviewer Fv7a -- rating 6  -- increased to7
* Reviewer d2Ug -- rating 4 -- increased to7

---

### Decision · Program_Chairs · 2026-01-26

Accept (Poster)